# TOWARDS OPTIMIZING TOP-$K$ RANKING METRICS IN RECOMMENDER SYSTEMS

## ABSTRACT

In the realm of recommender systems (RS), Top-$K$ metrics such as NDCG@$K$ are the gold standard for evaluating performance. Nonetheless, during the training of recommendation models, optimizing NDCG@$K$ poses significant challenges due to its inherent discontinuous nature and the intricacies of the Top-K truncation mechanism. Recent efforts to optimize NDCG@$K$ have either neglected the Top-$K$ truncation or suffered from low computational efficiency. To overcome these limitations, we propose SoftmaxLoss@$K$ (SL@$K$), a new loss function designed as a surrogate for optimizing NDCG@$K$ in RS. SL@$K$ integrates a quantile-based technique to handle the complex truncation term; and derives a smooth approximation of NDCG@$K$ to address discontinuity. Our theoretical analysis confirms the close bounded relationship between NDCG@$K$ and SL@$K$. Besides, SL@$K$ also exhibits several desirable properties including concise formulation, computational efficiency, and noisy robustness. Extensive experiments on four real-world datasets and three recommendation backbones demonstrate that SL@$K$ outperforms existing loss functions with a notable average improvement of 6.19%.

## 1 INTRODUCTION

Recommender systems (RS) (Ko et al., 2022; Zhang et al., 2019) have been widely applied in various personalized services (Nie et al., 2019; Ren et al., 2017). The primary goal of RS is to model users' preferences (scores) on items and subsequently retrieve a few items that users are most likely to interact with (Liu et al., 2009; Li et al., 2020; Hurley & Zhang, 2011). In practice, RS typically display only the Top-$K$ items to users. Therefore, *Top-K ranking metrics*, e.g., NDCG@$K$ (He et al., 2017b), are commonly used to evaluate recommendation performance. These metrics focuses on the quality of the items ranked within the Top-$K$ positions, as opposed to *full-ranking metrics* (e.g., NDCG) (Järvelin & Kekäläinen, 2017), which assess the entire ranking list.

Despite the widespread adoption of the NDCG@$K$ metric, optimizing this metric remains highly challenging: 1) The loss function is discontinuous and flat across most regions, rendering gradient-based optimization ineffective; 2) The loss computation involves truncating the ranking list, requiring the identification of whether an item appears in the Top-$K$ positions, which is difficult to manage.

Recent efforts have proposed *surrogate losses* (Lapin et al., 2016; 2017) to optimize NDCG@$K$, yet these approaches exhibit significant limitations:

- Some studies have focused on optimizing full-ranking metrics such as NDCG, without accounting for Top-$K$ truncation (Rashed et al., 2021; Chapelle & Wu, 2010; Taylor et al., 2008). A notable and successful example is the Softmax Loss (SL) (Wu et al., 2024a), which is easily implemented and serves as an upper bound for optimizing NDCG (Bruch et al., 2019). SL has been widely applied in practice and usually yield state-of-the-art (SOTA) performance (Wu et al., 2024b). However, NDCG and NDCG@$K$ are not always aligned — NDCG@$K$ focuses on the quality of a few top-ranked items, while NDCG evaluates the entire list. This discrepancy makes that optimizing NDCG does not always yield improvements in NDCG@$K$ and sometimes may even lead to performance degradation, as illustrated in Figure 1a.

- Other approaches have sought to optimize NDCG@$K$ by incorporating lambda weights (Burges et al., 2006; Wang et al., 2018) for each training instance in their LambdaLoss@$K$ (Jagerman et al., 2022). While this method has proven effective in document retrieval tasks (Liu et al., 2009),

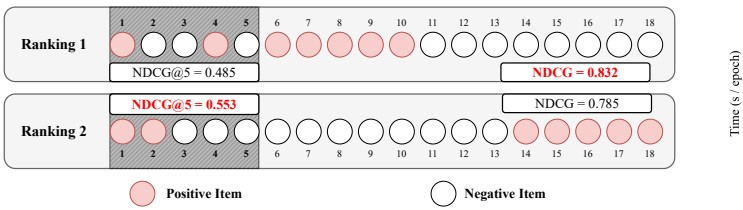
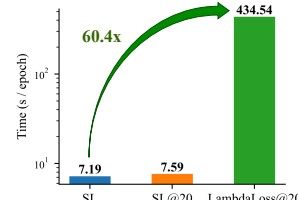

(a) Inconsistency between NDCG and NDCG@$K$.     (b) Execution time comparison.

Figure 1: (a) Illustration of inconsistency between NDCG and NDCG@$K$. Ranking 1 and Ranking 2 represent two different ranking schemes of the same set of items, where red/while circles denote positive/negative items respectively. While Ranking 1 has a better NDCG than Ranking 2, it has worse NDCG@5. (b) Execution time comparison (per epoch) on the Electronic dataset (8K items), where LambdaLoss@$K$ incurs a significantly higher computational overhead.

its application to RS remains impractical. The main challenge lies in efficiency: the calculation of lambda weights depends on the ranking positions of items, requiring a full sorting of items for each user at every iteration. This is computationally prohibitive in real-world RS given the immense number of users and items (cf. Figure 1b). Additionally, due to the sparsity of positive items in RS, most of lambda weights are extremely small (e.g., 99% are less than 0.005, cf. Appendix B), further hindering the effectiveness of the training process.

Given the critical importance of optimizing NDCG@$K$ and the limitations of existing approaches, there is a pressing need to develop a more effective surrogate loss for NDCG@$K$. In this work, we propose **SoftmaxLoss@$K$ (SL@$K$)**, incorporating the following strategies:

- To address the challenge of Top-$K$ truncation, we introduce a quantile-based technique (Koenker, 2005; Hao & Naiman, 2007; Shao, 2008). Specifically, we define a Top-$K$ quantile as a threshold score that separates the Top-$K$ items from the rest. This quantile can be efficiently estimated, and the complex top-$K$ truncation term can be reformulated as a simple comparison between an item's score and the quantile. This transformation makes the truncation both computationally efficient and tractable for optimization.
- To overcome the issue of discontinuity, we analyzes an upper bound for optimizing NDCG@$K$ and relax it into a fully continuous function. Our theoretical analysis proved that SL@$K$ serves as a tight upper bound for $-\log$ NDCG@$K$, ensuring both theoretical rigor and practical applicability.

Beyond its theoretical merits, SL@$K$ is concise in form and easy to implement. Compared to the conventional SL, SL@$K$ introduces only a quantile-based weight for each positive instance, which adds minimal computational overhead (cf. Figure 1b). Furthermore, our analysis reveals that SL@$K$ demonstrates enhanced robustness to false positive noise (Chen et al., 2023; Wang et al., 2021; Wen et al., 2019) — a common issue in RS, where some positive interactions may result from factors other than true user preference (e.g., misclicks).

To empirically validate the effectiveness of SL@$K$, we conduct extensive experiments across four real-world recommendation datasets using three typical recommendation backbones. The experimental results demonstrate that SL@$K$ achieves impressive performance improvements, with an average gain of 6.19% in NDCG@$K$. Additional experiments, including an exploration of the hyperparameter $K$ and robustness evaluations, confirm that SL@$K$ is not only well-aligned with NDCG@$K$ but also exhibits superior resistance to false positive noise.

## 2 PRELIMINARIES

### 2.1 TASK FORMULATION

This work focuses on Top-$K$ recommendation from implicit feedback, a widely-used scenario in recommender systems (RS) (Su, 2009; Zhu et al., 2019). Given a RS with a user set $\mathcal{U}$ and an item set $\mathcal{I}$, let $\mathcal{D} = \{y_{ui} : u \in \mathcal{U}, i \in \mathcal{I}\}$ denote the historical interactions between users and items, where $y_{ui} = 1$ indicates that user $u$ has interacted with item $i$, and $y_{ui} = 0$ indicates has not. For each user

$u$, we denote $\mathcal{P}_u = \{i \in \mathcal{I} : y_{ui} = 1\}$ as the set of positive items for $u$, and $\mathcal{N}_u = \mathcal{I} \setminus \mathcal{P}_u$ as the set of negative items. The recommendation task can be formulated as follows: learning user preference from $\mathcal{D}$ and recommending the Top-$K$ items that users are most likely to interact with.

Formally, modern RS typically infer user preferences for items with a learnable recommendation model $s_{ui} = f_\Theta(u, i)$, where $f_\Theta(u, i) : \mathcal{U} \times \mathcal{I} \to \mathbb{R}$ can be any flexible model architecture with parameters $\Theta$, mapping user/item features (e.g., IDs) into their preference scores $s_{ui}$. Subsequently, the Top-$K$ items with the highest $s_{ui}$ values are retrieved as recommendations. In this work, we focus not on model architecture design but instead on exploring the loss function. Given that the loss function guides the optimization direction of models, its importance cannot be overemphasized.

## 2.2 FORMULATION OF NDCG@$K$

Given the Top-$K$ recommendation nature of RS, Top-$K$ ranking metrics have been widely used to evaluate the recommendation performance. This work focuses on the most representative Top-$K$ ranking metric, NDCG@$K$ (Normalized Discounted Cumulative Gain with a Top-$K$ cutoff) (Järvelin & Kekäläinen, 2017). NDCG@$K$ not only measures the number of positive items within the Top-$K$ positions (as Recall@$K$ and Precision@$K$ do) but also considers their concrete ranking positions within the Top-$K$ ranking list (higher ranking with larger NDCG@$K$), which better reflects practical recommendation needs. Formally, NDCG@$K$ for each user can be formulated as follows:

$$\text{NDCG@}K(u) = \frac{\text{DCG@}K(u)}{\text{IDCG@}K(u)}, \quad \text{where } \text{DCG@}K(u) = \sum_{i \in \mathcal{P}_u} \frac{\mathbb{I}(\pi_{ui} \leq K)}{\log_2(\pi_{ui} + 1)} \quad (2.1)$$

where IDCG@$K$ is a normalizing constant representing the optimal DCG@$K$ value with an ideal ranking; $\mathbb{I}(\cdot)$ denotes indicator function; $\pi_{ui}$ denotes the ranking position of item $i$ for user $u$, which can be formally written as: $\pi_{ui} = \sum_{j \in \mathcal{I}} \mathbb{I}(s_{uj} \geq s_{ui})$.

**While NDCG@$K$ is widely applied, optimizing it presents significant challenges:**

- **Truncation Challenge:** The loss computation involves truncating the ranking list, i.e., the term $\mathbb{I}(\pi_{ui} \leq K)$, which requires identifying whether an item appears in the Top-$K$ positions. Efficient computation of this truncation is particularly challenging. Moreover, computing the gradient of this term for effective optimization remains an open problem.
- **Discontinuity Challenge:** The loss involves the computations of item ranking position $\pi_{ui}$, while $\pi_{ui}$ is a discontinuous function w.r.t. the model prediction scores $s_{ui}$. Moreover, the loss function is often flat over most regions (Bruch et al., 2019), making gradient-based optimization ineffective.

## 2.3 ANALYSES OVER EXISTING SURROGATE LOSS

To address these challenges, recent research has proposed surrogate losses for NDCG@$K$ optimization, but significant limitations remain. These approaches can be categorized into two types:

**Type 1: Optimizing NDCG without Top-$K$ truncation.** Some studies have focused on optimizing full-ranking metrics such as NDCG, without considering Top-$K$ truncation. NDCG optimization has been extensively explored, with approaches ranging from contrastive-based methods (e.g., Softmax Loss (Wu et al., 2024a)), ranking-based methods (e.g., Smooth-NDCG (Chapelle & Wu, 2010)), Gumbel-based methods (e.g., NeuralSort (Grover et al., 2019)), neural-based methods (e.g., GuidedRec (Rashed et al., 2021)). Among these methods, the most representative one is the **Softmax Loss (SL)** (Wu et al., 2024a), which has been widely used in practice and demonstrated effectiveness. Formally, SL is defined as:

$$\mathcal{L}_{\text{SL}}(u) = \sum_{i \in \mathcal{P}_u} \log \left( \sum_{j \in \mathcal{I}} \exp(d_{uij}/\tau) \right) \quad (2.2)$$

where $\tau$ is a temperature hyperparameter, and $d_{uij} = s_{uj} - s_{ui}$. SL offers multiple advantages: 1) **Theoretical guarantees**: SL has been proven to be an upper bound of $-\log \text{NDCG}$ (Bruch et al., 2019), ensuring that optimizing SL is consistent with optimizing NDCG, leading to SOTA performance. 2) **Efficiency**: SL has a concise form and does not require the computation of ranking positions, which is complex and time-consuming. Additionally, SL is compatible with negative

sampling — although its computation involves all items $j \in \mathcal{I}$, it can be efficiently accelerated through negative sampling (Wu et al., 2024b) or in-batch strategies (Wu et al., 2024a) during optimization.

While SL serves as an effective surrogate loss for NDCG, a gap remains between NDCG and NDCG@$K$, which limits its performance. As shown in Figure 1a, optimizing NDCG does not consistently improve NDCG@$K$ and sometimes even lead to performance drops. Thus, Top-$K$ truncation cannot be ignored and should be explicitly modeled during training.

**Type 2: Incorporating lambda weights.** Other researchers have proposed **Lambdaloss@$K$** (Jagerman et al., 2022), which optimizes NDCG@$K$ by incorporating lambda weights (Burges et al., 2006; Wang et al., 2018). In recommendation scenarios, Lambdaloss@$K$ can be written as:

$$\mathcal{L}_{\text{LambdaLoss}}(u) = \sum_{i \in \mathcal{P}_u, j \in \mathcal{N}_u} \mu_{uij} \cdot \text{Softplus}(d_{uij}) \tag{2.3}$$

where the lambda weight $\mu_{uij}$ is defined as

$$\mu_{uij} = \begin{cases} \eta_{uij} \cdot \left(1 - \dfrac{1}{\log_2(\max(\pi_{ui}, \pi_{uj}) + 1)}\right)^{-1} & \text{, if } \pi_{ui} > K \text{ or } \pi_{uj} > K \\ \eta_{uij} & \text{, else} \end{cases} \tag{2.4}$$

and

$$\eta_{uij} = \frac{1}{\log_2(|\pi_{ui} - \pi_{uj}| + 1)} - \frac{1}{\log_2(|\pi_{ui} - \pi_{uj}| + 2)} \tag{2.5}$$

Although Lambdaloss@$K$ has proven effective in document retrieval tasks, it is impractical for large-scale RS due to the following limitations:

- **High computational time cost.** The calculation of lambda weights $\mu_{uij}$ requires determining item ranking positions $\pi_{ui}$ and $\pi_{uj}$, which dynamically change during training. This necessitates a full sorting of items for each user at every iteration, with a complexity of $O(|\mathcal{U}||\mathcal{I}| \log |\mathcal{I}|)$, rendering it impractical for large-scale RS. While Monte Carlo sampling (Metropolis et al., 1953) could approximate rankings $\pi_{ui}$, its accuracy is questionable. More critically, The loss function is highly sensitive to estimation errors. Specifically, for instances where $\pi_{uj}$ is closer to $\pi_{ui}$, which have relatively larger $\mu_{uij}$ and contribute significantly to training, even small estimation errors can lead to substantial deviations. Our experiments show a performance degradation of over 30% when using sampling-based estimation in LambdaLoss@$K$ (cf. Table 3 and Appendix D.4).
- **Ineffective training due to extremely small lambda weights.** Due to the large item space and sparse positive instances in RS, most lambda weights $\mu_{uij}$ are extremely small since $|\pi_{ui} - \pi_{uj}|$ tends to be large. In our experiments, we found that 99% of weights are less than 0.005, suggesting that the gradients of Lambdaloss@$K$ are dominated by a few training instances, while others contribute negligibly (cf. Appendix B). This increases training instability and hampers model convergence. Furthermore, this issue complicates sampling estimation, as negative sampling exacerbates the problem: sampled instances often have small lambda weights, leading to gradient vanishing and consequently hindering training progress.

While optimizes NDCG@$K$ is promising, these limitations make Lambdaloss@$K$ less effective for RS. Developing a better NDCG@$K$ surrogate loss for recommendation warrants further exploration.

**Other Related Losses.** Beyond aforementioned losses, there are other conventional or advanced losses used in RS. For instance, BPR (Rendle et al., 2012) as one of the most classic approaches, approximately optimizes the AUC metric through pairwise comparisons. More recently, OPAUC (Dodd & Pepe, 2003) and LLPAUC (Shi et al., 2024) have been proposed to optimize partial AUC, with discussions on their theoretical relations with Recall@$K$ and Precision@$K$. However, their connections with NDCG@$K$ remain unknown. Additionally, these methods involve complex adversarial training, which may hinder their effectiveness and applicability. For a comprehensive overview of recent advancements in this area, readers are referred to Appendix A.

## 3 METHODOLOGY

In this section, we first introduce the proposed surrogate loss — **SoftmaxLoss@$K$ (SL@$K$)**, followed by a discussion of its properties. Finally, we detail the Top-$K$ quantile estimation method.

## 3.1 SOFTMAXLOSS@$K$: A SUPERIOR SURROGATE LOSS FOR NDCG@$K$

The primary challenges in optimizing NDCG@$K$ stem from the Top-$K$ truncation and the discontinuity. To address these challenges, we propose a novel surrogate loss, named SoftmaxLoss@$K$ (SL@$K$), leveraging the following strategies:

**Leveraging quantile technique.** The original truncation term $\mathbb{I}(\pi_{ui} \leq K)$ involves estimating the ranking position $\pi_{ui}$ and determining whether it is less than $K$, which is computationally difficult to handle efficiently. To overcome this, we introduce the Top-$K$ quantile $\beta_u^K$ of the preference scores for each user $u$, which is defined as:

$$\beta_u^K := \inf\{s_{ui} : \pi_{ui} \leq K\} \tag{3.1}$$

This quantile acts as a threshold score that separates the Top-$K$ items from the remainder. Specifically, if an item's score $s_{ui} \geq \beta_u^K$, it indicates that the item belongs to the Top-$K$ positions; conversely, $s_{ui} \leq \beta_u^K$ implies that it does not. Using this quantile, the truncation term can be simplified as:

$$\mathbb{I}(\pi_{ui} \leq K) = \mathbb{I}(s_{ui} \geq \beta_u^K) \tag{3.2}$$

This transformation reduces the problem to a simple comparison between the item's score $s_{ui}$ and the quantile $\beta_u^K$, thus avoiding the need to directly estimate the ranking position $\pi_{ui}$. This makes the Top-$K$ truncation both computationally efficient and easily optimizable.

Some may express concerns regarding the computational cost of estimating the Top-$K$ quantile. In fact, this quantile can be estimated efficiently and accurately using a sampling-based method with theoretical guarantees. We will discuss this in detail in Section 3.3.

**Deriving a continuous surrogate.** To tackle the discontinuity issue, we turn to relax NDCG@$K$ into a fully smooth function. Specifically, we aim to derive a smooth upper bound of $-\log \text{DCG@}K$, since optimizing this upper bound is equivalent to lifting NDCG@$K$[1]. To ensure well-definedness and rigor, we simply assume that DCG@$K$ is non-zero. In fact, this assumption is practical note that DCG@$K = 0$ is the worst result. During training, the scores of positive instances would be fast lifted and typically larger than those of negative instances. As a result, there is almost always at least one positive item in the Top-$K$ positions, ensuring that DCG@$K > 0$.

While several successful examples of relaxing (full-ranking) DCG exist as references (Bruch et al., 2019; Wang et al., 2018), special care must be taken to account for the differences in DCG@$K$ introduced by the truncation mechanism. We have the following relaxations for DCG@$K$:

$$-\log \text{DCG@}K(u) \overset{(3.2)}{=} -\log\left(\sum_{i \in \mathcal{P}_u} \mathbb{I}(s_{ui} \geq \beta_u^K) \frac{1}{\log_2(\pi_{ui} + 1)}\right) \tag{3.3a}$$

$$\overset{①}{\leq} -\log\left(\sum_{i \in \mathcal{P}_u} \mathbb{I}(s_{ui} \geq \beta_u^K) \frac{1}{\pi_{ui}}\right) \tag{3.3b}$$

$$= -\log\left(\sum_{i \in \mathcal{P}_u} \frac{\mathbb{I}(s_{ui} \geq \beta_u^K)}{H_u^K} \frac{1}{\pi_{ui}}\right) - \log H_u^K \tag{3.3c}$$

$$\overset{②}{\leq} \sum_{i \in \mathcal{P}_u} \frac{\mathbb{I}(s_{ui} \geq \beta_u^K)}{H_u^K} \left(-\log \frac{1}{\pi_{ui}}\right) - \log H_u^K \tag{3.3d}$$

$$\overset{③}{\leq} \sum_{i \in \mathcal{P}_u} \mathbb{I}(s_{ui} \geq \beta_u^K) \log \pi_{ui} \tag{3.3e}$$

where $H_u^K = \sum_{v \in \mathcal{P}_u} \mathbb{I}(s_{uv} \geq \beta_u^K)$, denoting the number of positive instances in Top-K positions (a.k.a. Top-K hits) for user $u$. Equation (3.3c) is well-defined and $H_u^K \geq 1$ due to our non-zero assumption[2]. Several important relaxations are applied in Equation (3.3): ① is due to $\log_2(\pi_{ui} + 1) \leq \pi_{ui}$; ② is due to Jensen's inequality (Jensen, 1906); ④ is due to $H_u^K \geq 1$.

---

[1]Note that optimizing DCG@$K$ and NDCG@$K$ is equivalent, as the normalization term IDCG is a constant.
[2]Due to the assumption that DCG@$K > 0$, there is at least one Top-$K$ hit $i$ such that $s_{ui} \geq \beta_u^K$.

The motivation behind the relaxations ① and ② is to manage the complexity of the fractional term $1/\log_2(\pi_{ui} + 1)$, which involves the ranking position $\pi_{ui}$ in the denominator. By transforming the fractional term into a more concise form, we simplify the calculation. This transformation helps to avoid numerical instability and better supports sampling-based estimation. Similar techniques have been employed in Softmax Loss (SL) (Wu et al., 2024a; Bruch et al., 2019) to handle NDCG. For the relaxation ③, we drop the term $H_u^K$ due to its computational complexity. While retaining this term could potentially lead to improved performance, we empirically find that the gains are marginal, whereas the additional computational overhead is significant.

We can express indicator function with Heaviside step function $\delta(x) = \mathbb{I}(x \geq 0)$, and express the the ranking position $\pi_{ui}$ based on the scores $s_{ui}$, i.e., $\pi_{ui} = \sum_{j \in \mathcal{I}} \mathbb{I}(s_{uj} \geq s_{ui}) = \sum_{j \in \mathcal{I}} \delta(d_{uij})$, where $d_{uij} = s_{uj} - s_{ui}$. Thus, Equation (3.3e) can be re-written as:

$$(3.3\text{e}) = \sum_{i \in \mathcal{P}_u} \delta(s_{ui} - \beta_u^K) \cdot \log \left( \sum_{j \in \mathcal{I}} \delta(d_{uij}) \right) \tag{3.4}$$

To further address the discontinuity of the Heaviside functions $\delta(\cdot)$ in Equation (3.4), we approximate them by two continuous activations $\sigma_w$ and $\sigma_d$, resulting in the following **SoftmaxLoss@$K$ (SL@$K$)**:

$$\mathcal{L}_{\text{SL@}K}(u) = \sum_{i \in \mathcal{P}_u} \underbrace{\sigma_w(s_{ui} - \beta_u^K)}_{\text{weight: } w_{ui}} \cdot \underbrace{\log \left( \sum_{j \in \mathcal{I}} \sigma_d(d_{uij}) \right)}_{\text{SL term: } \mathcal{L}_{\text{SL}}(u,i)} \tag{3.5}$$

Note that exponential and sigmoid are two conventional activation functions to approximate the Heaviside function $\delta(\cdot)$ — exponential are employed by SL, and sigmoid has been shown to provide a tighter approximation. Here we recommend using two different activations: $\sigma_d$ as the exponential with $\sigma_d(x) = e^{x/\tau_d}$, and $\sigma_w$ as the sigmoid with $\sigma_w(x) = 1/(1 + e^{-x/\tau_w})$, where $\tau_d$ and $\tau_w$ denote temperature hyperparameters. This configuration ensures that SL@$K$ serves as a tight upper bound for $-\log \text{DCG@}K$ (cf. Theorem 3.1). In contrast, if both activations are chosen as sigmoid, the bound relations do not hold; if both are chosen as exponential, the bound is not as tight as in our setting. Readers may refer to the discussions in Appendix C.1 for further details.

## 3.2 ANALYSES OF SOFTMAXLOSS@$K$

Our proposed SoftmaxLoss@$K$ (SL@$K$) offers several advantages:

**Concise and efficient.** The proposed SL@$K$ has a concise form (3.5). Compared to conventional SL, SL@$K$ only introduces an additional quantile-based weight $w_{ui}$ for each instance, which just involves a simple difference between the scores $s_{ui}$ and the quantiles $\beta_u^K$. SL@$K$ inherits the benefits of SL, while the introduction of $w_{ui}$ can be intuitively understood: it assigns larger weights to positive instances with higher scores $s_{ui}$, emphasizing those within the Top-$K$ positions during optimization. This aligns with the principles of Top-$K$ ranking metric for recommendation.

The introduction of $w_{ui}$ does not incur significantly computational overhead. The quantile estimation and weight calculation in SL@$K$ are efficient and do not require the time-consuming estimation of ranking positions, as in LambdaLoss@$K$. Moreover, similar to SL, SL@$K$ supports negative sampling, leading to further acceleration during training.

The time complexity of SL@$K$ changes from $\mathcal{O}(|\mathcal{U}|\bar{P}N)$ of SL to $\mathcal{O}(|\mathcal{U}|\bar{P}N + |\mathcal{U}|N \log N)$, where $\bar{P}$ denotes the average number of positive items per user; and $N$ denotes the size of sampled negative items satisfying $N \ll |\mathcal{I}|$. The additional complexity $\mathcal{O}(|\mathcal{U}|N \log N)$ arises from the quantile estimation (cf. Section 3.3), which remains efficient, as $\log N$ is typically smaller than $\bar{P}$. Our experiments also confirm the computational efficiency of SL@$K$ (cf. Table 3 in Section 4.2).

**Theoretical guarantees.** We establish theoretical connections between SL@$K$ and NDCG@$K$:

**Theorem 3.1** (SL@$K$ as a surrogate loss for NDCG@$K$). *For any user $u$, if the Top-K hits $H_u^K > 1$, then SL@$K$ serves as an upper bound of $-\log \text{DCG@}K$, i.e.,*

$$-\log \text{DCG@}K(u) \leq \mathcal{L}_{\text{SL@}K}(u) \tag{3.6}$$

*when $H_u^K = 1$, a slightly looser but effective bound holds, i.e., $-\frac{1}{2}\log \text{DCG@}K(u) \leq \mathcal{L}_{\text{SL@}K}(u)$.*

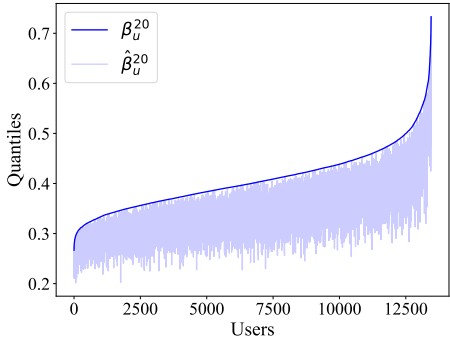 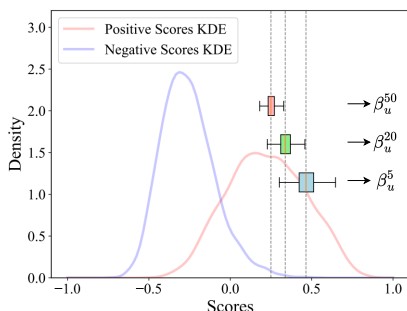

(a) Ideal quantile $\beta_u^K$ vs. estimated quantile $\hat{\beta}_u^K$.    (b) Distributions of Top-$K$ quantile $\beta_u^K$ and scores.

Figure 2: Illustration of the estimated quantile $\hat{\beta}_u^{20}$ compared with the ideal quantile $\beta_u^{20}$ across users on the Electronic dataset, where users are sorted by $\beta_u^{20}$. The estimation error is $0.06 \pm 0.03$. (b) The values of the ideal quantiles, compared with the distributions of positive scores $s_{ui}$ and negative scores $s_{uj}$, using Kernel Density Estimation (KDE) (Parzen, 1962) to illustrate the distribution.

The proof is presented in Appendix C.2. From Equation (3.4), the derivation is straightforward, except for the careful handling of the activation functions. The assumptions of $H_u^K > 1$ is commonly satisfied in practice, as the training process tends to increase the scores of positive items, making them typically larger than those of negative items (cf. Appendix C.2 for empirical validation). These theoretical properties guarantee the effectiveness of SL@$K$ — minimizing SL@$K$ is equivalent to maximizing DCG@$K$, leading to recommendation performance improvements.

**Robustness to false positive noise.** False positive instances (Chen et al., 2023) are prevalent in recommendation systems, arising from various factors such as iclckbait (Wang et al., 2021), item position bias (Hofmann et al., 2014), or accidental interactions (Adamopoulos & Tuzhilin, 2014). Recent studies have shown that such noise can significantly mislead model training and degrade performance (Wen et al., 2019). Interestingly, the introduction of the weight $w_{ui}$ in SL@$K$ helps mitigate this issue. False positives, which often resemble negative instances, tend to have lower prediction scores $s_{ui}$ than true positives. As a result, they receive smaller weights $w_{ui}$ and contribute less in model training, which enhances the robustness of model, as analyzed in Appendix C.3.

### 3.3 Top-$K$ Quantile Estimation

Quantile estimation has been extensively studied in the field of statistics (Koenker, 2005; Hao & Naiman, 2007; Bickel & Doksum, 2015). In this work, we develop a simple Monte Carlo sampling-based strategy (Metropolis et al., 1953). The approach is straightforward: for each user, we randomly sample a small set of $N$ items and estimate the Top-$K$ quantile from this sampled set. The complexity of this method is $\mathcal{O}(|\mathcal{U}|N \log N)$, as it only requires sorting the items in the sample set. Despite its simplicity, this method comes with theoretical guarantees:

**Theorem 3.2** (Sample quantile estimation error). *For any c.d.f. $F$ and any $p \in (0,1)$ , the p-th quantile[3] is define as $\theta_p := F^{-1}(p) = \inf\{t : F(t) \geq p\}$. We sample $N$ samples $\{X_i\}_{i=1}^{N} \stackrel{i.i.d.}{\sim} F$ , suppose that $F_N(t) = \frac{1}{N}\sum_{i=1}^{N} \mathbb{I}(X_i \leq t)$ is the empirical c.d.f. , and the p-th estimated quantile is defined as $\hat{\theta}_p := F_N^{-1}(p)$. Then, for any $\epsilon > 0$, we have*

$$\Pr\left(\left|\hat{\theta}_p - \theta_p\right| > \varepsilon\right) \leq 4e^{-2N\delta_\varepsilon^2} \tag{3.7}$$

*where $\delta_\varepsilon = \min\{F(\theta_p + \varepsilon) - p, p - F(\theta_p - \varepsilon)\}$.*

The proof is provided in Appendix D.1. Theorem 3.2 provides theoretical foundation of sampling-based estimation that the error between the estimated quantile and the ideal quantile is bounded by a function that decreases exponentially with the sample size $N$. This implies that the Top-$K$ quantile $\beta_u^K$ can be estimated with arbitrary precision provided a sufficiently large $N$.

---

[3]Here we adopt the definition of $p$-th quantile to generalize the theory to the continuous case. In the context of RS, this can be simply interpreted as the Top-$(p \cdot |\mathcal{I}|)$ quantile.

In practice, this simple strategy can be further improved by leveraging the properties of recommendation systems. As shown in Figure 2b, the scores of positive items are typically much higher than those of negative items, and the Top-$K$ quantile is often located within the range of positive item scores. Therefore, it is more effective to retain all positive instances and randomly sample a small set of negative instances for quantile estimation. This strategy, though simple, yields more accurate results. Figure 2a provides an example of estimated quantiles across users on the Electronic dataset, with a sample size of $N = 1000$. The estimated quantile $\hat{\beta}_u^{20}$ closely matches the optimal $\beta_u^{20}$, with an average deviation of only 0.06. More examples and details can refer to Appendix D.2.

## 4 EXPERIMENTS

### 4.1 EXPERIMENTAL SETUP

**Datasets and backbones.** To ensure fair comparisons, our experimental setup closely follows Wu et al. (2024a;b)'s prior work. We conduct experiments on four widely-used datasets: Health, Electronic, Gowalla, and Book. Additionally, given the inefficiency of LambdaLoss@$K$ in handling these large datasets, we further evaluate its performance on two additional datasets with relatively small scale, Movielens and Food. Detailed descriptions of the datasets can be found in Appendix F.1.

We also evaluate the proposed losses using three distinct recommendation backbones: the classic Matrix Factorization (MF) model (Koren et al., 2009), the representative graph-based model LightGCN (He et al., 2020), and the SOTA method XsimGCL (Yu et al., 2023).

**Compared losses.** We compare our SL@$K$ loss with the following conventional or SOTA losses: 1) the classic **BPR** (Rendle et al., 2012); 2) the SOTA **Softmax Loss (SL)** (Wu et al., 2024a) and its DRO-enhanced variants (Shapiro, 2017) including **AdvInfoNCE** (Zhang et al., 2024) and **BSL** (Wu et al., 2024b); 3) model-based NDCG surrogate loss **GuidedRec** (Rashed et al., 2021); 4) **LambdaLoss@$K$** (Jagerman et al., 2022) that optimizes NDCG@$K$; 5) **LLPAUC** (Shi et al., 2024) that optimizes partial AUC metric. The readers may refer to Appendix F.4 for more details.

**Hyperparameters settings.** For fair comparisons, SL@$K$ sets the temperature $\tau_d$ (cf. Equation (3.5)) to be the same as the optimal $\tau$ in SL (cf. Equation (2.2)), and uses the same negative sampling as SL for sample quantile estimation and training, with the negative sampling number $N = 1000$. The implementation details can be found in Appendix F.4, and the optimal hyperparameters of these losses are reported in Appendix F.6.

### 4.2 ANALYSES ON EXPERIMENTS RESULTS

**SL@$K$ vs. Existing losses.** Table 1 presents the performance comparison of SL@$K$ against existing losses. As shown, SL@$K$ consistently outperforms all competing losses across various datasets and backbones. The improvements are substantial, with an average increase of 6.19%. This highlights the importance of explicitly modeling Top-$K$ truncation during optimization, which cannot be overlooked. Since SL@$K$ is more closely aligned with the NDCG@$K$ metric, we observe its superiority over existing losses. Interestingly, SL@$K$ also demonstrates strong performance on Recall@$K$ metric. This can be attributed to the fact that optimizing NDCG@$K$ naturally increases the number of positive items in the Top-$K$ positions, thereby enhancing Recall@$K$ performance.

**Performance comparison with varying $K$.** Table 2 illustrates the performance across different values of $K$. We observe that SL@$K$ consistently outperforms the compared methods for various values of $K$. However, as $K$ increases, the magnitude of the improvements decreases. This observation aligns with our intuition. Specifically, the truncation mechanism has a greater impact when $K$ is small. As $K$ increases, the Top-$K$ metric NDCG@$K$ degrades to the full-ranking metric NDCG. Consequently, the advantage of optimizing for NDCG@$K$ relatively diminishes as $K$ grows.

**SL@$K$ vs. Lambdaloss@$K$.** We further compare SL@$K$ with Lambdaloss@$K$ on two relatively small datasets, with the results presented in Table 3. Although both losses are designed to optimize NDCG@$K$, our experiments show that SL@$K$ consistently outperforms Lambdaloss@$K$. This performance gap can primarily be attributed to the extremely skewed lambda weights in Lambdaloss@$K$, which hinder its training effectiveness. Moreover, we observe that Lambdaloss@$K$ incurs signifi-

Table 1: Performance comparison of SL@$K$ with existing losses. The best results are highlighted in bold, and the best baselines are underlined. "Imp." denotes the improvement of SL@$K$ over the best baseline; "R@20" denotes the metric Recall@20; and "D@20" denotes the metric NDCG@20.

| Backbone | Loss | Health | | Electronic | | Gowalla | | Book | |
|---|---|---|---|---|---|---|---|---|---|
| | | R@20 | D@20 | R@20 | D@20 | R@20 | D@20 | R@20 | D@20 |
| MF | BPR | 0.1575 | 0.1209 | 0.0816 | 0.0527 | 0.1355 | 0.1111 | 0.0665 | 0.0453 |
| | GuidedRec | 0.1573 | 0.1084 | 0.0644 | 0.0385 | 0.1135 | 0.0863 | 0.0518 | 0.0361 |
| | LLPAUC | 0.1671 | 0.1219 | 0.0821 | 0.0499 | 0.1610 | 0.1189 | 0.1150 | 0.0811 |
| | SL | 0.1737 | 0.1264 | 0.0821 | 0.0529 | 0.2064 | 0.1624 | 0.1559 | 0.1210 |
| | AdvInfoNCE | 0.1660 | 0.1236 | 0.0829 | 0.0527 | 0.2067 | 0.1627 | 0.1557 | 0.1172 |
| | BSL | 0.1737 | 0.1264 | 0.0834 | 0.0530 | 0.2071 | 0.1630 | 0.1563 | 0.1212 |
| | **SL@20** | **0.1804** | **0.1373** | **0.0892** | **0.0587** | **0.2121** | **0.1709** | **0.1612** | **0.1269** |
| | **Imp. %** | **+3.86%** | **+8.62%** | **+6.95%** | **+10.75%** | **+2.41%** | **+4.85%** | **+3.13%** | **+4.70%** |
| LightGCN | BPR | 0.1618 | 0.1203 | 0.0813 | 0.0524 | 0.1745 | 0.1402 | 0.0984 | 0.0678 |
| | GuidedRec | 0.1550 | 0.1073 | 0.0657 | 0.0393 | 0.0921 | 0.0686 | 0.0468 | 0.0310 |
| | LLPAUC | 0.1685 | 0.1207 | 0.0831 | 0.0507 | 0.1616 | 0.1192 | 0.1147 | 0.0810 |
| | SL | 0.1691 | 0.1235 | 0.0823 | 0.0526 | 0.2068 | 0.1628 | 0.1567 | 0.1220 |
| | AdvInfoNCE | 0.1706 | 0.1264 | 0.0823 | 0.0528 | 0.2066 | 0.1625 | 0.1568 | 0.1177 |
| | BSL | 0.1691 | 0.1236 | 0.0823 | 0.0526 | 0.2069 | 0.1628 | 0.1568 | 0.1220 |
| | **SL@20** | **0.1791** | **0.1369** | **0.0894** | **0.0587** | **0.2128** | **0.1729** | **0.1625** | **0.1280** |
| | **Imp. %** | **+4.98%** | **+8.31%** | **+7.58%** | **+11.17%** | **+2.85%** | **+6.20%** | **+3.64%** | **+4.92%** |
| XSimGCL | BPR | 0.1496 | 0.1108 | 0.0777 | 0.0508 | 0.1966 | 0.1570 | 0.1269 | 0.0905 |
| | GuidedRec | 0.1539 | 0.1088 | 0.0760 | 0.0473 | 0.1685 | 0.1277 | 0.1275 | 0.0951 |
| | LLPAUC | 0.1519 | 0.1083 | 0.0781 | 0.0481 | 0.1632 | 0.1200 | 0.1363 | 0.1008 |
| | SL | 0.1534 | 0.1113 | 0.0772 | 0.0490 | 0.2005 | 0.1570 | 0.1549 | 0.1207 |
| | AdvInfoNCE | 0.1499 | 0.1072 | 0.0776 | 0.0489 | 0.2010 | 0.1564 | 0.1568 | 0.1179 |
| | BSL | 0.1649 | 0.1201 | 0.0800 | 0.0507 | 0.2037 | 0.1597 | 0.1550 | 0.1207 |
| | **SL@20** | **0.1718** | **0.1322** | **0.0860** | **0.0569** | **0.2095** | **0.1717** | **0.1624** | **0.1277** |
| | **Imp. %** | **+4.18%** | **+10.07%** | **+7.50%** | **+12.01%** | **+2.85%** | **+7.51%** | **+3.57%** | **+5.80%** |

Table 2: Performance comparisons with varying $K$ on Health and Electronic datasets and MF backbone. The best results are highlighted in bold, and the best baselines are underlined. "Imp." denotes the improvement of SL@$K$ over the best baseline; "D@20" denotes the metric NDCG@20.

| Health | D@5 | D@20 | D@50 |
|---|---|---|---|
| BPR | 0.0934 | 0.1209 | 0.1602 |
| GuidedRec | 0.0771 | 0.1084 | 0.1477 |
| LLPAUC | 0.0909 | 0.1219 | 0.1575 |
| SL | 0.0921 | 0.1264 | 0.1611 |
| AdvInfoNCE | 0.0918 | 0.1236 | 0.1607 |
| BSL | 0.0921 | 0.1264 | 0.1611 |
| **SL@$K$** | **0.1072** | **0.1373** | **0.1733** |
| **Imp. %** | **+14.78%** | **+8.62%** | **+7.57%** |

| Electronic | D@5 | D@20 | D@50 |
|---|---|---|---|
| BPR | 0.0347 | 0.0527 | 0.0699 |
| GuidedRec | 0.0225 | 0.0385 | 0.0546 |
| LLPAUC | 0.0305 | 0.0499 | 0.0687 |
| SL | 0.0352 | 0.0529 | 0.0696 |
| AdvInfoNCE | 0.0340 | 0.0527 | 0.0695 |
| BSL | 0.0345 | 0.0530 | 0.0695 |
| **SL@$K$** | **0.0401** | **0.0587** | **0.0760** |
| **Imp. %** | **+13.92%** | **+10.75%** | **+8.73%** |

Table 3: Performance comparison of SL@$K$ with the Lambdaloss@$K$ on MF backbone. "Imp." denotes the improvement of SL@$K$ over LambdaLoss@$K$, while "Degr." denotes the degradation of LambdaLoss@$K$ caused by the sample estimation. The average running time per epoch is reported.

| Loss | Movielens | | | Food | | |
|---|---|---|---|---|---|---|
| | Recall@20 | NDCG@20 | Time (s) | Recall@20 | NDCG@20 | Time (s) |
| LambdaLoss@20 | 0.3418 | 0.3466 | 26 | 0.0530 | 0.0382 | 494 |
| LambdaLoss@20 (Sample) | 0.1580 | 0.1603 | 6 | 0.0335 | 0.0238 | 36 |
| Degr. % (Sample) | -53.77% | -53.75% | N/A | -36.79% | -37.70% | N/A |
| **SL@20** | **0.3580** | **0.3677** | **2** | **0.0635** | **0.0465** | **8** |
| **Imp. %** | **+4.53%** | **+6.09%** | N/A | **+19.81%** | **+21.73%** | N/A |

Figure 3: NDCG@20 performance of SL@$K$ compared with SL under varying ratios of imposed false positive instances. "Imp." indicates the improvement of SL@$K$ over SL.

Table 4: Performance exploration of SL@$K$ on NDCG@$K$ with inconsistent $K$.

| Health | D@5 | D@20 | D@50 |
|---|---|---|---|
| SL@5 | **0.1072** | 0.1363 | 0.1723 |
| SL@20 | 0.1067 | **0.1373** | 0.1728 |
| SL@50 | 0.1065 | 0.1365 | **0.1733** |

| Electronic | D@5 | D@20 | D@50 |
|---|---|---|---|
| SL@5 | **0.0401** | 0.0585 | 0.0756 |
| SL@20 | 0.0401 | **0.0587** | 0.0758 |
| SL@50 | 0.0400 | 0.0586 | **0.0760** |

cantly higher computational costs compared to SL@$K$. While sampling strategies could be employed to accelerate Lambdaloss@$K$, they lead to substantial (over 30%) performance degradation.

**Noise Robustness Study.** In Figure 3, we assess robustness of SL@$K$ to false positive instances. Following (Wu et al., 2024b), we manually introduce a certain ratio of negative instances as noisy positive instances during training. As shown in Figure 3, as the noise ratio increases, SL@$K$ demonstrates greater improvements over SL, indicating that SL@$K$ exhibits superior robustness to false positive noise. This finding is consistent with our analysis in Section 3.2.

**Consistency Exploration of NDCG@$K$ and SL@$K$.** Table 4 presents the performance of NDCG@$K$ and SL@$K$ for varying values of $K$ in $\{5, 20, 50\}$. We observe that the best performance is achieved when the value of $K$ in SL@$K$ matches that of NDCG@K. This result aligns with our expectations. Specifically, when the value of $K$ in SL@$K$ differs from that in NDCG@$K$, e.g., SL@20 for NDCG@50, where SL@20 would target at optimizing NDCG@20 rather than NDCG@50, such discrepancy leads to a performance drop.

**Exploration of Hyperparameter $\tau_w$.** Figure 4 depicts the model performance with varying $\tau_w$. Initially, performance improves as $\tau_w$ increases, but beyond a certain point, further increases lead to a decline in performance. This behavior reflects an inherent trade-off. When $\tau_w$ is small, the surrogate for NDCG@$K$ is tighter, potentially improving alignment with the target metric but increasing the training difficulty due to the decrease in Lipschitz smoothness. Conversely, as $\tau_w$ increases, the approximation would be loose, also impacting model performance.

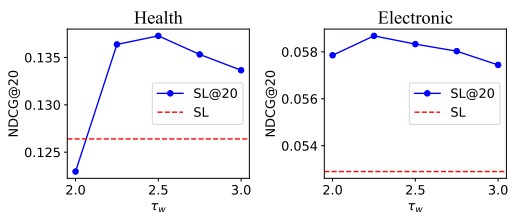

Figure 4: Sensitivity analysis of SL@$K$ on $\tau_w$.

## 5 CONCLUSION AND FUTURE DIRECTIONS

This work introduces a novel loss function, SoftmaxLoss@$K$ (SL@$K$), designed for optimizing NDCG@$K$. SL@$K$ leverages a quantile-based technique to handle the truncation challenge and derives a smooth approximation to tackle the discontinuity problem. Our theoretical analysis confirms the close bounded relationship between NDCG@$K$ and SL@$K$. Beyond its theoretical strengths, SL@$K$ offers a concise formulation, introducing only quantile-based weights on top of the conventional Softmax Loss, making it both easy to implement and computationally efficient.

Looking ahead, a promising direction for future work would be the development of incremental quantile estimation methods, which could further enhance the efficiency of SL@$K$ and support the incremental learning of recommendation models. Additionally, investigating the application of SL@$K$ in other domains would be valuable, as Top-$K$ metrics are widely utilized in tasks such as multimedia retrieval, question answering, link prediction, and anomaly detection.

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

## A    RELATED WORK

**Recommendation models.** As a fundamental component of recommender systems, recommendation models aim to predict the user-item interactions. One of the most popular paradigms is collaborative filtering (CF) (Su, 2009; Zhu et al., 2019). CF-based models assume that users with similar preferences will have similar interactions with items. Therefore, a common practice to implement CF models is to parameterize the user and item embeddings and predict the interactions by the vector similarity between user and item embeddings.

The earliest works stem from the idea of Matrix Factorization (MF) (Koren et al., 2009), which factorizes the user-item interaction matrix into user and item embedding vectors, such as MF (Koren et al., 2009), SVD (Deerwester et al., 1990; Bell et al., 2007), SVD++ (Koren, 2008), NCF (He et al., 2017a), etc. However, MF-based models have limitations in capturing high-order relations, since they only consider the first-order interactions. To address this issue, some works have proposed to incorporate the graph structure of user-item interactions, using Graph Neural Networks (GNNs) (Wu et al., 2022; Kipf & Welling, 2016; Wang et al., 2019). GNN-based models, such as LightGCN (He et al., 2020), NGCF (Wang et al., 2019), and APDA (Zhou et al., 2023), have achieved great success in recommendation. Moreover, the most recent works, including SGL (Wu et al., 2021) and XSimGCL (Yu et al., 2023), introduce contrastive learning (Liu et al., 2021; Oord et al., 2018) for graph data augmentation, achieving state-of-the-art performance in recommendation.

**Recommendation losses.** Recommendation loss, which significantly impacts the effectiveness of recommendation models, is gaining increasing attention from researchers in the field. The earliest works treat recommendation as a simple regression or binary classification problem, utilizing pointwise losses such as MSE (He & Chua, 2017) and BCE (He et al., 2017a). However, due to neglecting the ranking essence in recommendation, these pointwise losses usually result in inferior recommendation performance.

To address the limitations of pointwise losses, pairwise losses such as BPR (Rendle et al., 2012) have been proposed. BPR aims to learn a partial order between positive and negative items, which is a surrogate loss for AUC metric and achieves significant improvements over pointwise losses. Following BPR, listwise losses (Cao et al., 2007) such as Softmax Loss (SL) (Wu et al., 2024a) extends the pairwise ranking to listwise, i.e., maximizing the likelihood of the entire list of items consisting of one positive item and multiple negative items. SL has been proven as a NDCG surrogate loss and achieves state-of-the-art performance in recommendation (Wu et al., 2024a; Bruch et al., 2019).

Given the success of ranking losses, recent works have attempted to further improve ranking performance from different perspectives. For instance, some works have proposed to further improve the robustness of SL by introducing Distributional Robust Optimization (DRO) (Shapiro, 2017), e.g., AdvInfoNCE (Zhang et al., 2024) and BSL (Wu et al., 2024b). Other works try to directly optimize the ranking metrics including NDCG (Järvelin & Kekäläinen, 2017) and MRR (Lu et al., 2023). Among them, LambdaRank (Burges et al., 2006) and LambdaLoss (Wang et al., 2018) are the most representative works, which serve as the NDCG surrogate losses with a different form compared to SL. There are also some works focusing on optimizing NDCG from other approaches, e.g., GuidedRec (Rashed et al., 2021) uses neural networks, Smooth-NDCG (Chapelle & Wu, 2010) designs a smooth ranking position indicator, SoftNDCG (Taylor et al., 2008) considers the rank distribution, NeuralSort (Grover et al., 2019) leverages Gumbel-Softmax trick for optimization, etc.

Despite the success of the aforementioned ranking losses, they still have limitations in practice, as real-world recommender systems only retrieve a small subset of items for users, i.e., Top-$K$ recommendation (Li et al., 2020; Hurley & Zhang, 2011). The Top-$K$ ranking metrics (e.g., NDCG@$K$), which consider solely the top-ranked items, could be inconsistent with the full ranking metrics (e.g., NDCG). Therefore, the NDCG surrogate losses like SL and LambdaLoss may obtain suboptimal performance in practical recommendation scenarios. To address this issue, directly optimizing the Top-$K$ ranking metrics has become increasingly important.

Several existing works focus on Top-$K$ metrics optimization. For example, LLPAUC (Shi et al., 2024) optimizes the lower-left part of AUC, which is a surrogate loss for Recall@$K$ and Precision@$K$. Prec@$K$ (Lu et al., 2019) directly optimize the Precision@$K$ in deep image embedding task. LambdaLoss@$K$ (Jagerman et al., 2022), which is a reweighted LambdaLoss, achieves a NDCG@$K$

surrogate loss in document retrieval tasks. However, LLPAUC and Prec@$K$ are not designed for optimizing NDCG@$K$. Besides, LLPAUC involves complex adversarial training, hinders its effectiveness and applicable. Moreover, Prec@$K$ and LambdaLoss@$K$ are not specifically designed for recommendation, would suffer from serious inefficiency issue when transferred to recommendation scenarios. The skewed lambda weight in LambdaLoss@$K$ also hinders its effective training. Therefore, it is still an open problem to design an efficient and effective surrogate loss for optimizing NDCG@K in recommendation.

# B ANALYSIS OF LAMBDA WEIGHT IN LAMBDALOSS@$K$

In this section, we provide a detailed analysis of the lambda weight $\mu_{uij}$ in LambdaLoss@$K$ (Jagerman et al., 2022), which is defined as

$$\mu_{uij} = \begin{cases} \eta_{uij} \cdot \left(1 - \dfrac{1}{\log_2(\max(\pi_{ui}, \pi_{uj}) + 1)}\right)^{-1} & \text{, if } \pi_{ui} > K \text{ or } \pi_{uj} > K \\ \eta_{uij} & \text{, else} \end{cases} \tag{2.4}$$

and

$$\eta_{uij} = \frac{1}{\log_2(|\pi_{ui} - \pi_{uj}| + 1)} - \frac{1}{\log_2(|\pi_{ui} - \pi_{uj}| + 2)} \tag{2.5}$$

Since $\eta_{uij}$ is the difference between the reciprocals of adjacent discount terms $1/\log_2(\cdot)$, this causes the lambda weight $\mu_{uij}$ to rapidly approach 0 when $|\pi_{ui} - \pi_{uj}|$ is large, i.e., when the ranking positions of the two items differ significantly. This indicates that during training, only negative items that are close to positive items receive sufficient gradients, while most negative items do not get effective trained. In fact, this is counter-intuitive and leads to inefficient training.

The following Figure B.1 shows the lambda weight $\mu_{uij}$ of Top-20 items in LambdaLoss@5, with a minimum value of 0.005. Even with a ranking difference less than 20, $\mu_{uij}$ is nearly vanishing. This means that in a RS with $|\mathcal{I}|$ items, the lambda weight $\delta_{ui}$ has at most $40|\mathcal{I}|$ values greater than 0.005, which is less than 1% of the total number of items in the practical RS with usually more than 4K items. This clearly indicates the gradient vanishing issue in LambdaLoss@$K$. Conversely, there are a certain ratio ($1/|\mathcal{I}|$) of the lambda weights are greater than 0.3, which dominate the gradients and have a decisive impact on the optimization direction, which increases training instability and hampers model convergence. This also indicates we can not use a large learning rate to mitigate issue of gradient vanishing during sampling estimation. As the few instances with large lambda weights could be sampled occasionally and lead to numerical explosion if we use a large learning rate. Overall, the extreme long-tail distribution of lambda weights makes optimization challenging and cannot be easily resolved by simply adjusting the learning rate.

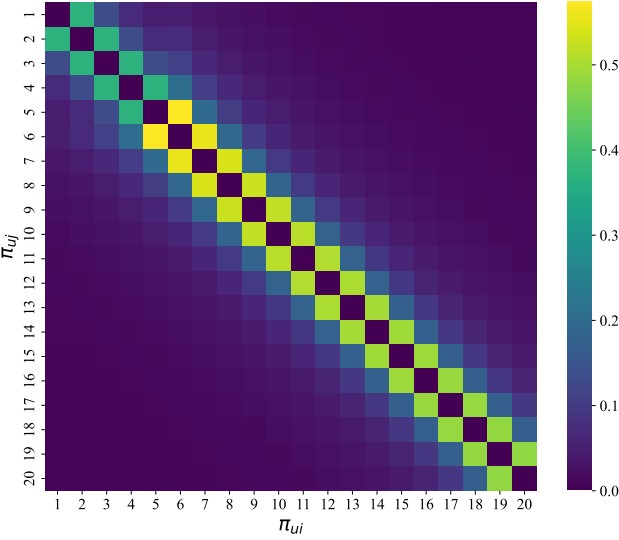

Figure B.1: The lambda weight $\mu_{uij}$ of Top-20 items in LambdaLoss@5.

# C  ADDITIONAL ANALYSIS OF SL@$K$

## C.1  DISCUSSION ON THE ACTIVATION FUNCTIONS IN SL@$K$

In Equation (3.5), we smooth SL@$K$ by two conventional activation functions, i.e., the sigmoid function $\sigma_w(x) = 1/(1 + \exp(-x/\tau_w))$ and the exponential function $\sigma_d(x) = \exp(x/\tau_d)$, where $\tau_w$ and $\tau_d$ are the temperature parameters. In this section, we will discuss the rationale behind the selection of these activation functions, as summarized in Table C.1.

Table C.1: Comparison of different activation functions choices in SL@$K$.

| $(\sigma_w, \sigma_d)$ | Sigmoid | Exponential |
|---|---|---|
| **Sigmoid** **Exponential** | ✗ (not achieve upper bound) ✗ (not achieve upper bound) | ✓ **(Our SL@$K$ loss)** ✗ (not tight enough) |

**Case 1:** $(\sigma_w, \sigma_d) = $ **(Sigmoid, Sigmoid).** To achieve an upper bound of DCG@$K$ from Equation (3.4) to Equation (3.5), since $\sigma_w(\cdot) \geq 0$ whether $\sigma_w(\cdot)$ chooses the sigmoid or exponential function, the positivity of $\mathcal{L}_{\mathrm{SL}}(u,i) = \log\left(\sum_{j\in\mathcal{I}} \sigma_d(d_{uij})\right)$ should be guaranteed. However, if we choose the sigmoid function for $\sigma_d(\cdot)$, this positivity may not be guaranteed, and thus leads to a failure to achieve a surrogate loss with theoretical guarantees. Moreover, given that the sigmoid function is not an upper bound of $\delta(\cdot)$, choosing the sigmoid function for $\sigma_w(\cdot)$ would also fail to achieve the upper bound of DCG@$K$.

**Case 2:** $(\sigma_w, \sigma_d) = $ **(Sigmoid, Exponential).** This is our proposed SL@$K$ loss, which achieves a tight upper bound for $-\log \mathrm{DCG}@K$, as proven in Theorem 3.1 and Appendix C.2.

**Case 3:** $(\sigma_w, \sigma_d) = $ **(Exponential, Sigmoid).** Similar to Case 1, the sigmoid function could make the $\mathcal{L}_{\mathrm{SL}}(u,i)$ term not positive and thus fail to achieve the upper bound of DCG@$K$.

**Case 4:** $(\sigma_w, \sigma_d) = $ **(Exponential, Exponential).** In this case, SL@$K$ indeed serves as an upper bound of $-\log \mathrm{DCG}@K$, but the exponential function is not tight enough to approximate the Heaviside step function $\delta(\cdot)$, leading to a loose upper bound. In fact, the difference between the sigmoid function $1/(1 + \exp(-x/\tau_w))$ and $\delta(x)$ is $1/(1 + \exp(|x|/\tau_w)) \approx \exp(-|x|/\tau_w)$ when $\tau_w$ is small. In contrast, the difference between the exponential function $\exp(x/\tau_d)$ and $\delta(x)$ is $\exp(x/\tau_d) - 1 \approx x/\tau_d$ when $x > 0$ and $\tau_d$ is large. It's obvious that the sigmoid function is a better approximation of the Heaviside step function. Additionally, even though the sigmoid function does not serve as an upper bound of $\delta(\cdot)$, it can still be used in SL@$K$ to surrogate DCG@$K$ with tighter upper bound, as proven in Theorem 3.1.

## C.2  PROOF OF THEOREM 3.1

**Theorem C.1** (Theorem 3.1, SL@$K$ as a surrogate loss for NDCG@$K$). *For any user $u$, if the Top-K hits $H_u^K > 1$, then SL@$K$ serves as an upper bound of $-\log \mathrm{DCG}@K$, i.e.,*

$$-\log \mathrm{DCG}@K(u) \leq \mathcal{L}_{\mathrm{SL}@K}(u) \tag{3.6}$$

*when $H_u^K = 1$, a slightly looser but effective bound holds, i.e., $-\frac{1}{2}\log \mathrm{DCG}@K(u) \leq \mathcal{L}_{\mathrm{SL}@K}(u)$.*

*Proof of Theorem 3.1.* Recall that in Section 3.1, we derive Equation (3.3d), i.e.,

$$-\log \mathrm{DCG}@K(u) \leq \sum_{i\in\mathcal{P}_u} \frac{\mathbb{I}(s_{ui} \geq \beta_u^K)}{H_u^K} \log \pi_{ui} - \log H_u^K \tag{C.1}$$

By the assumption of $H_u^K \geq 1$, the last term $-\log H_u^K$ can be relaxed, resulting in

$$-\log \mathrm{DCG}@K(u) \leq \sum_{i\in\mathcal{P}_u} \frac{\mathbb{I}(s_{ui} \geq \beta_u^K)}{H_u^K} \log \pi_{ui} \tag{C.2}$$

Recall again that

$$\pi_{ui} = \sum_{j\in\mathcal{I}} \mathbb{I}(s_{uj} \geq s_{ui}) = \sum_{j\in\mathcal{I}} \delta(d_{uij}) \leq \sum_{j\in\mathcal{I}} \sigma_d(d_{uij}) \tag{C.3}$$

where $d_{uij} = s_{uj} - s_{ui}$, $\delta(x) = \mathbb{I}(x \geq 0)$ is the Heaviside step function, and $\sigma_d(x) = \exp(x/\tau_d)$ is the exponential function serving as a smooth upper bound of $\delta(x)$ for any $x$ and $\tau_d > 0$. Therefore, Equation (C.2) can be further relaxed as

$$-\log \text{DCG@}K(u) \leq \sum_{i \in \mathcal{P}_u} \frac{1}{H_u^K} \delta(s_{ui} - \beta_u^K) \log \left( \sum_{j \in \mathcal{I}} \sigma_d(d_{uij}) \right) \qquad \text{(C.4)}$$

**Case 1.** In the case of $H_u^K > 1$, we have

$$\frac{1}{H_u^K} \delta(s_{ui} - \beta_u^K) \leq \frac{1}{2} \delta(s_{ui} - \beta_u^K) \leq \sigma_w(s_{ui} - \beta_u^K) \qquad \text{(C.5)}$$

where $\sigma_w(x) = 1/(1 + \exp(-x/\tau_w))$ is the sigmoid function with temperature $\tau_w > 0$. The last inequality in Equation (C.5) holds due to $\sigma_w(s_{ui} - \beta_u^K) \geq \frac{1}{2}$ if $s_{ui} > \beta_u^K$. Therefore, by Equations (C.4) and (C.5), we have

$$-\log \text{DCG@}K(u) \leq \sum_{i \in \mathcal{P}_u} \sigma_w(s_{ui} - \beta_u^K) \log \left( \sum_{j \in \mathcal{I}} \sigma_d(d_{uij}) \right) \qquad \text{(C.6)}$$

which exactly corresponds to the SL@$K$ loss $\mathcal{L}_{\text{SL@}K}(u)$ in Equation (3.5). Therefore, SL@$K$ serves as an upper bound of $-\log \text{DCG@}K$ when $H_u^K > 1$.

**Case 2.** In the case of $H_u^K = 1$, there only exists one positive item $i^* \in \mathcal{P}_u$ with $s_{ui^*} \geq \beta_u^K$. In this case, Equation (C.1) can be reduced to

$$-\log \text{DCG@}K(u) \leq \log \pi_{ui^*} \leq \log \left( \sum_{j \in \mathcal{I}} \sigma_d(d_{ui^*j}) \right) \qquad \text{(C.7)}$$

Since $s_{ui^*} \geq \beta_u^K$, we have $\sigma_w(s_{ui^*} - \beta_u^K) \geq \frac{1}{2}$, which leads to

$$-\frac{1}{2} \log \text{DCG@}K(u) \leq \sigma_w(s_{ui^*} - \beta_u^K) \log \left( \sum_{j \in \mathcal{I}} \sigma_d(d_{ui^*j}) \right) \leq \mathcal{L}_{\text{SL@}K}(u) \qquad \text{(C.8)}$$

This completes the proof. $\qquad \square$

**Discussion.** The condition in Theorem 3.1 is easy to satisfy in practice. For example, on Electronic dataset, SL@20 achieves $H_u^{20} > 1$ for 53.32%, 81.92%, and 95.66% of users within 5, 10, and 20 epochs, respectively.

## C.3  Gradient Analysis and False Positive Denoising

SL@$K$ inherently possesses the denoising ability to resist the false positive noise (e.g., misclicks), which is common in RS (Wen et al., 2019). To theoretically analyze the denoising ability of SL@$K$, we conduct a gradient analysis as follows:

$$\nabla_{\mathbf{u}} \mathcal{L}_{\text{SL@}K} = \mathbb{E}_{i \sim \mathcal{P}_u} \left[ w_{ui} \nabla_{\mathbf{u}} \mathcal{L}_{\text{SL}}(u, i) + \frac{1}{\tau_w} w_{ui}(1 - w_{ui}) \mathcal{L}_{\text{SL}}(u, i) \nabla_{\mathbf{u}} s_{ui} \right] \qquad \text{(C.9)}$$

Therefore, we can derive an upper bound of $\|\nabla_{\mathbf{u}} \mathcal{L}_{\text{SL@}K}\|$ as

$$\|\nabla_{\mathbf{u}} \mathcal{L}_{\text{SL@}K}\| \leq \mathbb{E}_{i \sim \mathcal{P}_u} \left[ w_{ui} \left( \|\nabla_{\mathbf{u}} \mathcal{L}_{\text{SL}}(u, i)\| + \frac{1}{\tau_w} \mathcal{L}_{\text{SL}}(u, i) \|\nabla_{\mathbf{u}} s_{ui}\| \right) \right] \qquad \text{(C.10)}$$

It's evident that the above gradient upper bound of SL@$K$ w.r.t. the user embedding $\mathbf{u}$ is controlled by the weight $w_{ui}$. For any false positive item $i$ with low score $s_{ui}$, $w_{ui}$ will be sufficiently small, which reduces its impact on the gradient. This analysis indicates that SL@$K$ is robust to false positive noise, highlighting its denoising ability.

# D SAMPLE QUANTILE ESTIMATION

## D.1 SAMPLE QUANTILE ESTIMATION ERROR BOUND

In this section, we provide the proof of Theorem 3.2.

**Theorem D.1** (Theorem 3.2, Sample quantile estimation error). *For any c.d.f. $F$ and any $p \in (0, 1)$, the $p$-th quantile is define as $\theta_p := F^{-1}(p) = \inf\{t : F(t) \geq p\}$. We sample $N$ samples $\{X_i\}_{i=1}^{N} \overset{i.i.d.}{\sim} F$, suppose that $F_N(t) = \frac{1}{N}\sum_{i=1}^{N} \mathbb{I}(X_i \leq t)$ is the empirical c.d.f., and the $p$-th estimated quantile is defined as $\hat{\theta}_p := F_N^{-1}(p)$. Then, for any $\epsilon > 0$, we have*

$$\Pr\left(\left|\hat{\theta}_p - \theta_p\right| > \varepsilon\right) \leq 4e^{-2N\delta_\varepsilon^2} \tag{3.7}$$

*where $\delta_\varepsilon = \min\{F(\theta_p + \varepsilon) - p, p - F(\theta_p - \varepsilon)\}$.*

To proof Theorem 3.2, we first introduce the following lemma.

**Lemma D.2** (Dvoretzky-Kiefer-Wolfowitz (DKW) inequality (Massart, 1990; Bickel & Doksum, 2015)). *For any c.d.f. $F$ and the corresponding empirical c.d.f. $F_N$, given the sup-norm distance between $F_N$ and $F$ defined as $\|F_N - F\|_\infty = \sup_{t \in \mathbb{R}}\{|F_N(t) - F(t)|\}$, we have*

$$\Pr\left(\|F_N - F\|_\infty > \varepsilon\right) \leq 2e^{-2N\varepsilon^2} \tag{D.1}$$

The estimation error bound of the sample quantile technique (cf. Theorem 3.2) can be simply derived from the DKW inequality (cf. Lemma D.2) as follows.

*Proof of Theorem 3.2.* Consider the error between $\hat{\theta}_p$ and $\theta_p$, we have

$$\begin{aligned}
\Pr(\hat{\theta}_p > \theta_p + \varepsilon) &= \Pr(p > F_N(\theta_p + \varepsilon)) \\
&= \Pr(F(\theta_p + \varepsilon) - F_N(\theta_p + \varepsilon) > F(\theta_p + \varepsilon) - p) \\
&\leq \Pr(\|F_N - F\|_\infty > \delta_\varepsilon^+)
\end{aligned} \tag{D.2}$$

where $\delta_\varepsilon^+ = F(\theta_p + \varepsilon) - p$. Analogously, let $\delta_\varepsilon^- = p - F(\theta_p - \varepsilon)$, we have

$$\Pr(\hat{\theta}_p < \theta_p - \varepsilon) \leq \Pr(\|F_N - F\|_\infty > \delta_\varepsilon^-) \tag{D.3}$$

Therefore, we have the two side error bound (cf. Equation (3.7)) by setting $\delta_\varepsilon = \min\{\delta_\varepsilon^+, \delta_\varepsilon^-\}$, which completes the proof. □

## D.2 SAMPLE QUANTILE ESTIMATION TRICKS FOR RECOMMENDATION

In Section 3.3, we introduce a sampling trick to estimate the Top-$K$ quantile $\beta_u^K$ in RS. Specifically, our sampled items will include all positive items $\mathcal{P}_u$ and $N$ ($\ll I$) sampled negative items $\hat{\mathcal{N}}_u = \{j_k \overset{i.i.d.}{\sim} \mathcal{N}_u\}_{k=1}^{N}$. Since the Top-$K$ quantile is usually located within the score range of positive items, this trick can estimate the quantile more effectively than directly i.i.d. sampling from all items, as shown in Figure D.2.

However, applying this sampling trick leads to a theoretical gap. Since the sampled items $\hat{\mathcal{I}}_u = \mathcal{P}_u \cup \hat{\mathcal{N}}_u$ are not i.i.d. sampled from the whole item set $\mathcal{I}$, we should not directly sample the $(K/I)$-th quantile of $\hat{\mathcal{I}}_u$ as the estimated quantile $\hat{\beta}_u^K$, which may introduce serious bias. Instead, under a reasonable assumption that all Top-$\min(K, P_u)$ items are positive items, we should set the estimated quantile $\hat{\beta}_u^K$ as:

- If $K \leq P_u$, $\hat{\beta}_u^K$ should be set as the Top-$K$ score of $\{s_{ui}\}$, where $i \in \mathcal{P}_u$.
- If $K > P_u$, $\hat{\beta}_u^K$ should be set as the $((K - P_u)/I)$-th quantile of $\{s_{uj}\}$, where $j \in \hat{\mathcal{N}}_u$.

The sampling trick above can be seen as non-bias. Nevertheless, this sampling setting is still not practical in RS. In the case of $K > P_u$, the quantile ratio $(K - P_u)/I$ can be too small and even less than $1/N$ (e.g., $K = 20, I = 10^5, N = 10^3$). Therefore, the estimated quantile $\hat{\beta}_u^K$ could be theoretically higher than all the negative item scores and can not be estimated by sampling $\hat{\mathcal{N}}_u$.

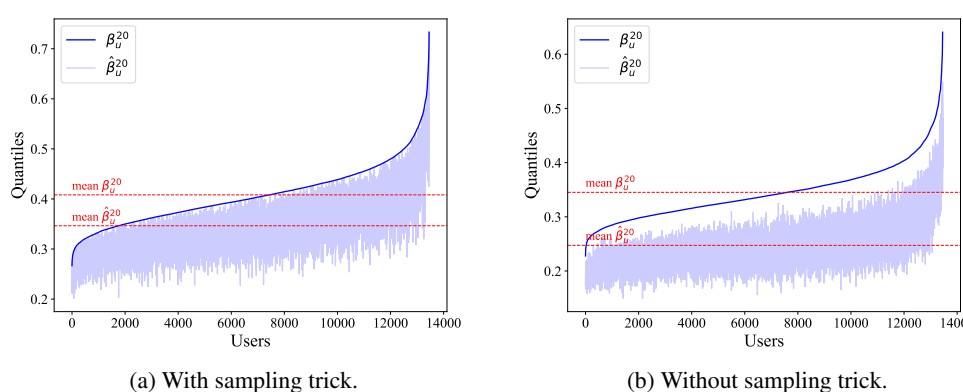

(a) With sampling trick.

(b) Without sampling trick.

Figure D.2: Comparison of sample quantile estimation with and without the sampling trick for recommendation, using the same setting as Figure 2.

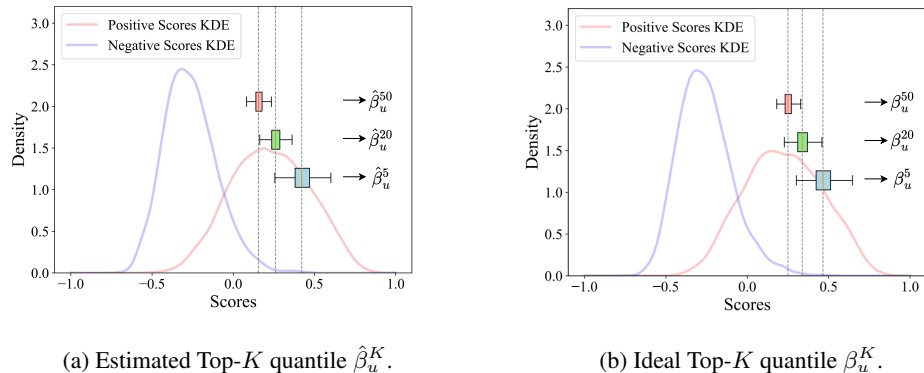

(a) Estimated Top-$K$ quantile $\hat{\beta}_u^K$.

(b) Ideal Top-$K$ quantile $\beta_u^K$.

Figure D.3: Comparison of the estimated Top-$K$ quantile $\hat{\beta}_u^K$ with the ideal Top-$K$ quantile $\beta_u^K$, using the same setting as Figure 2.

Given the impracticability of the above non-bias sampling setting, we slightly modify the sampling trick. Specifically, we set $\hat{\beta}_u^K$ as the Top-$K$ score of $\{s_{uk}\}$, where $k \in \mathcal{P}_u \cup \hat{\mathcal{N}}_u$. This sampling trick perfectly fits the above non-bias case when $K \leq P_u$. In the case of $K > P_u$, this setting actually estimates the $(K - P_u)/N$-th quantile of negative item scores, introducing a slight bias but also making the training more stable. Moreover, it's clear that the estimated quantile $\hat{\beta}_u^K$ will always be lower than the ideal Top-$K$ quantile $\beta_u^K$ under this sampling trick (cf. Figure D.2), which leads to a more moderate truncation in training SL@$K$, as shown in Figure D.3.

### D.3 QUANTILE REGRESSION

Quantile regression method (Koenker, 2005; Hao & Naiman, 2007) can also be used for sample quantile estimation. Specifically, to estimate the $p$-th quantile, the quantile regression loss can be defined as

$$\mathcal{L}_{\text{QR}}(u) = \mathbb{E}_{i \sim \mathcal{I}} \left[ (1 - p)(s_{ui} - \hat{\beta}_u)_+ + p(\hat{\beta}_u - s_{ui})_+ \right] \tag{D.4}$$

or equivalently

$$\mathcal{L}_{\text{QR}}(u) = \mathbb{E}_{i \sim \mathcal{I}} \left[ (s_{ui} - \hat{\beta}_u)(\delta(s_{ui} - \hat{\beta}_u) - p) \right] \tag{D.5}$$

where $(\cdot)_+ = \max(\cdot, 0)$, $\hat{\beta}_u$ is the estimated $p$-th quantile, and note that $x \cdot \delta(x) = x_+, x_+ - (-x)_+ = x$, for any $x \in \mathbb{R}$.

Suppose that $S$ is a random variable representing the score of items $s_{ui}$, and $F_S$ is the c.d.f. of $S$ on $\mathbb{R}$. Since $i \sim \mathcal{I}$ means that $i$ follows the uniform distribution on $\mathcal{I}$, we can rewrite the quantile regression loss in Equation (D.4) as

$$
\begin{aligned}
\mathcal{L}_{\text{QR}}(u) &= \mathbb{E}_{S \sim F_S} \left[ (1-p)(S - \hat{\beta}_u)_+ + p(\hat{\beta}_u - S)_+ \right] \\
&= \int_{-\infty}^{\hat{\beta}_u} p(\hat{\beta}_u - S) \mathrm{d}F_S(S) + \int_{\hat{\beta}_u}^{\infty} (1-p)(S - \hat{\beta}_u) \mathrm{d}F_S(S)
\end{aligned}
\tag{D.6}
$$

Let $\beta_u = \arg\min_{\hat{\beta}_u} \mathcal{L}_{\text{QR}}(u)$, we have

$$
p \int_{-\infty}^{\beta_u} \mathrm{d}F_S(S) = (1-p) \int_{\beta_u}^{\infty} \mathrm{d}F_S(S)
\tag{D.7}
$$

resulting $\int_{\beta_u}^{\infty} \mathrm{d}F_S(S) = p$, i.e., the optimal $\hat{\beta}_u$ is precisely the $p$-th quantile of scores $S$.

This regression-based approach can reduce the complexity of SL@$K$ to $\mathcal{O}(PN)$ with $N$ negative sampling. However, in practice, it is found that training quantile regression is relatively difficult to control, so we still adopt the above sampling trick in Appendix D.2.

## D.4 SAMPLE RANKING ESTIMATION

Similar to sample quantile estimation, sample ranking estimation can also be applied to estimate the ranking position $\pi_{ui}$. Specifically, we can sample $N$ negative items $\hat{\mathcal{N}}_u = \{j_k \overset{\text{i.i.d.}}{\sim} \mathcal{N}_u\}_{k=1}^N$, and sort the sampled items $i \in \hat{\mathcal{I}}_u = \mathcal{P}_u \cup \hat{\mathcal{N}}_u$ by scores $\{s_{ui}\}$. Then, for any item $i$, given the sample ranking position $\pi_{ui}^*$ in the sampled items $\hat{\mathcal{I}}_u$, the estimated ranking position $\hat{\pi}_{ui}$ in the entire item set is rescaled as

$$
\hat{\pi}_{ui} = \pi_{ui}^* \cdot \frac{|\mathcal{I}|}{|\hat{\mathcal{I}}_u|}
\tag{D.8}
$$

Compared to sample quantile estimation, sample ranking estimation may result in greater errors, primarily because the estimated ranking $\hat{\pi}_{ui}$ obtained from sample ranking estimation is always fixed, i.e., $1, 1 + |\mathcal{I}|/|\hat{\mathcal{I}}_u|, 1 + 2|\mathcal{I}|/|\hat{\mathcal{I}}_u|, \cdots$. Obviously, sample ranking estimation will result in an expected error of at least $\frac{1}{2}|\mathcal{I}|/|\hat{\mathcal{I}}_u| \approx \frac{1}{2}|\mathcal{I}|/N$, which decreases inversely w.r.t. $N$. However, the error in sample quantile estimation decreases exponentially w.r.t. $N$, leading to better estimation accuracy. Therefore, sample ranking estimation is not suitable for losses that are extremely sensitive to ranking positions, such as LambdaLoss (Wang et al., 2018) and LambdaLoss@$K$ (Jagerman et al., 2022), as discussed in Appendix B.

# E  SL@$K$ OPTIMIZATION

In this section, we provide the detailed optimization algorithm of SL@$K$ (cf. Equation (3.5)) in Algorithm E.1, which is based on the sample quantile estimation trick in Appendix D.2.

In practical SL@$K$ optimization, to mitigate the training difficulties caused by frequent changes in quantiles due to score variations (especially in the early stages), we introduce a quantile update interval hyperparameter $T_\beta$, i.e., updating the quantiles every $T_\beta$ epochs.

---

**Algorithm E.1** SL@$K$ optimization

---

**Input:** user and item sets $\mathcal{U}, \mathcal{I}$; dataset $\mathcal{D} = \{y_{ui} \in \{0,1\} : u \in \mathcal{U}, i \in \mathcal{I}\}$; score function $s_{ui} : \mathcal{U} \times \mathcal{I} \to \mathbb{R}$ with parameters $\Theta$; negative sampling number $N$; the number of epochs $T$; the number of $K$; temperature parameters $\tau_w, \tau_d$; quantile update interval $T_\beta$.

1: Initialize the estimated Top-$K$ quantiles $\hat{\beta}_u^K \leftarrow 0$ for all $u \in \mathcal{U}$.
2: **for** $t = 1, 2, \ldots, T$ **do**
3:      **for** $u \in \mathcal{U}$ **do**
4:          Let $\mathcal{P}_u = \{i : y_{ui} = 1\}$ be the positive items of user $u$.
5:          Let $\mathcal{N}_u = \{i : y_{ui} = 0\}$ be the negative items of user $u$.

$\triangleright$ Estimate the quantiles $\hat{\beta}_u^K$
$\triangleright$ Complexity: $\mathcal{O}((|\mathcal{P}_u| + N)\log(|\mathcal{P}_u| + N))$
$\triangleright$ Complexity: $\approx \mathcal{O}(N \log N)$

6:          **if** $t \equiv 0 \mod T_\beta$ **then**
7:              Sample $N$ negative items $\hat{\mathcal{N}}_u = \{j_k \overset{\text{i.i.d.}}{\sim} \mathcal{N}_u\}_{k=1}^N$, let $\hat{\mathcal{I}}_u = \mathcal{P}_u \cup \hat{\mathcal{N}}_u$.
8:              Sort items $\hat{i} \in \hat{\mathcal{I}}_u$ by scores $\{s_{u\hat{i}}\}$.
9:              Estimate the Top-$K$ quantile $\hat{\beta}_u^K \leftarrow \hat{\mathcal{I}}_u[K]$, i.e., the $K$-th top-ranked item in $\hat{\mathcal{I}}_u$.
10:          **end if**

$\triangleright$ Optimize $\Theta$ by SL@$K$ loss
$\triangleright$ Complexity: $\mathcal{O}(|\mathcal{P}_u|N)$

11:          Sample $N$ negative items $\hat{\mathcal{N}}_u = \{j_k \overset{\text{i.i.d.}}{\sim} \mathcal{N}_u\}_{k=1}^N$.
12:          **for** $i \in \mathcal{P}_u$ **do**
13:              Compute the weight $w_{ui} = \sigma_w(s_{ui} - \hat{\beta}_u^K)$, where $\sigma_w = \sigma(\cdot/\tau_w)$.
14:              Compute the SL loss $\mathcal{L}_{\text{SL}}(u, i) = \log \sum_{j \in \hat{\mathcal{N}}_u} \sigma_d(d_{uij})$, where $\sigma_d = \exp(\cdot/\tau_d)$.
15:          **end for**
16:          Compute the loss $\mathcal{L}_{\text{SL}@K}(u) = \sum_{i \in \mathcal{P}_u} w_{ui} \cdot \mathcal{L}_{\text{SL}}(u, i)$.
17:          Update the parameters $\Theta$ by minimizing $\mathcal{L}_{\text{SL}@K}(u)$.
18:      **end for**
19: **end for**
**Output:** the optimized parameters $\Theta$.

---

# F  EXPERIMENTAL DETAILS

## F.1  DATASETS

In our experiments, we adopt six benchmark datasets summarized in Table F.2:

- **Health / Electronic / Book** (He & McAuley, 2016a; McAuley et al., 2015): These datasets are collected from the Amazon dataset, a large crawl of product reviews from Amazon[4]. The 2014 version of Amazon dataset contains 142.8 million reviews spanning May 1996 to July 2014.
- **Gowalla** (Cho et al., 2011): The Gowalla dataset is a check-in dataset collected from the location-based social network Gowalla[5], including 1M users, 1M locations, and 6M check-ins.
- **Movielens** (Harper & Konstan, 2015): The Movielens dataset is a movie rating dataset collected from Movielens[6]. We use the Movielens-100K version, which contains 100,000 ratings from 1000 users on 1700 movies.
- **Food** (Majumder et al., 2019): The Food dataset consists of 180K recipes and 700K recipe reviews covering 18 years of user interactions and uploads on Food.com[7].

Table F.2: Statistics of the datasets.

| Dataset | #Users | #Items | #Interactions | Density |
|---|---|---|---|---|
| Health (He & McAuley, 2016a) | 1,974 | 1,200 | 48,189 | 0.02034 |
| Electronic (He & McAuley, 2016a) | 13,455 | 8,360 | 234,521 | 0.00208 |
| Gowalla (Cho et al., 2011) | 29,858 | 40,988 | 1,027,464 | 0.00084 |
| Book (He & McAuley, 2016a) | 135,109 | 115,172 | 4,042,382 | 0.00026 |
| Movielens (Harper & Konstan, 2015) | 939 | 1,016 | 80,393 | 0.08427 |
| Food (Majumder et al., 2019) | 5,875 | 9,852 | 233,038 | 0.00403 |

In dataset preprocessing, following the standard practice in Wang et al. (2019), we use a 10-core setting (He & McAuley, 2016b), i.e. all users and items have at least 10 interactions. To remove the low-quality interactions, we only retain the interactions with ratings greater or equal to 3 (if available). After preprocessing, we randomly split the datasets into 80% training and 20% test sets, and a 10% validation set is further randomly split from the training set for hyperparameter tuning.

## F.2  RECOMMENDATION SCENARIOS

In this paper, we evaluate the performance of each method mainly under the following two Top-$K$ recommendation scenarios:

- **IID scenario** (He et al., 2020): The IID scenario is the most common recommendation scenario, where the training and test sets are i.i.d. split from the whole dataset and have the same distributions. We closely follow the setting in He et al. (2020).
- **False Positive Noise scenario** (Wu et al., 2024b): The Noise scenario is widely adopted to evaluate the denoising capabilities. Our false positive noise setting is similar to the false negative noise setting in Wu et al. (2024b). Specifically, for each user $u$, we randomly sample $\lceil r \times P_u \rceil$ negative items and flip them to positive items as false positive noise. The range of noise ratios $r$ is $\{5\%, 10\%, 15\%, 20\%\}$.

## F.3  RECOMMENDATION BACKBONES

Recommendation backbones, or the recommendation models, are the core components of RS. In the scope of this paper, the recommendation backbones can be seen as the score function $s_{ui} : \mathcal{U} \times \mathcal{I} \to \mathbb{R}$ with parameters $\Theta$. It is crucial to evaluate the effectiveness of the recommendation loss on different backbones to ensure their generalization and consistency.

---

[4]https://www.amazon.com/
[5]https://en.wikipedia.org/wiki/Gowalla
[6]https://movielens.org/
[7]https://www.food.com/

In our experiments, we implement three popular recommendation backbones:

- **MF** (Koren et al., 2009): MF is the most basic but still effective recommendation model, which factorizes the user-item interaction matrix into user and item embeddings. All the embedding-based recommendation models use MF as the first layer. Specifically, we set the embedding size $d = 64$ for all settings, following the setting in Wang et al. (2019).

- **LightGCN** (He et al., 2020): LightGCN is a effective GNN-based recommendation model. LightGCN performs graph convolution on the user-item interaction graph, so as to aggregate the high-order interactions. Specifically, LightGCN simplifies NGCF (Wang et al., 2019) and only retains the non-parameterized graph convolution. In our experiments, we set the number of layers as 2, which aligns with the original setting in He et al. (2020).

- **XSimGCL** (Yu et al., 2023): XSimGCL is a novel recommendation model based on contrastive learning (Jaiswal et al., 2020; Liu et al., 2021). Based on a 3-layers LightGCN, XSimGCL adds a random noise to the output embeddings of each layer, and introduces the contrastive learning between the final layer and the $l^*$-th layer, i.e. adding a auxiliary InfoNCE (Oord et al., 2018) loss between these two layers. Following the original Yu et al. (2023)'s setting, the modulus of random noise between each layer is set as 0.1, the contrastive layer $l^*$ is set as 1 (where the embedding layer is 0-th layer), the temperature of InfoNCE is set as 0.1, and the weight of the auxiliary InfoNCE loss is searching from $\{0.05, 0.1, 0.2\}$.

### F.4 COMPARED METHODS AND HYPERPARAMETERS SETTING

To adequately evaluate the effectiveness of SL@$K$, we reproduce the following SOTA recommendation losses and search for the optimal hyperparameters using grid search. In loss optimization, we use Adam (Kingma & Ba, 2014) optimizer with learning rate as lr, and weight decay ($L_2$ regularization hyperparameter) as wd. The batch size is set as 1024, and the number of epochs is set as 200. If the negative sampling is needed, we set the negative sampling number $N = 1000$, except for the Movielens dataset, which is set to 200 due to the smaller number of items.

- **BPR** (Rendle et al., 2012): A pairwise loss based on the Bayesian Maximum Likelihood Estimation (MLE) (Casella & Berger, 2024). The objective of BPR is to learn a partial order of the items, i.e., the positive items should be ranked higher than the negative items. Furthermore, BPR is a surrogate loss for AUC metric (Rendle et al., 2012; Silveira et al., 2019).
  - **Hyperparameters**: lr $\in \{10^{-1}, 10^{-2}, 10^{-3}, 10^{-4}\}$, wd $\in \{0, 10^{-4}, 10^{-5}, 10^{-6}\}$.
  - **Score function** $s_{ui}$: dot product.

- **GuidedRec** (Rashed et al., 2021): A BCE (He et al., 2017a) loss with DCG surrogate learning guidance. GuidedRec is not a DCG surrogate loss. Instead, it learns a surrogate loss model to estimate DCG. During training, GuidedRec maximizes the estimated DCG while minimizing the MSE (He & Chua, 2017) between the estimated DCG and the real DCG.
  - **Hyperparameters**: lr $\in \{10^{-1}, 10^{-2}, 10^{-3}\}$, wd $\in \{0, 10^{-4}, 10^{-5}, 10^{-6}\}$.
  - **Score function** $s_{ui}$: cosine similarity.

- **LLPAUC** (Shi et al., 2024): A surrogate loss for lower-left part of AUC. LLPAUC has been shown as a surrogate loss for metrics such as Recall@$K$ and Precision@$K$ (Fayyaz et al., 2020).
  - **Hyperparameters**: lr $\in \{10^{-1}, 10^{-2}, 10^{-3}\}$, wd $\in \{0, 10^{-4}, 10^{-5}, 10^{-6}\}$, hyperparameters $\alpha \in \{0.1, 0.3, 0.5, 0.7, 0.9\}$ and $\beta \in \{0.01, 0.1\}$, which follows Shi et al. (2024)'s setting.
  - **Score function** $s_{ui}$: cosine similarity.

- **Softmax Loss (SL)** (Wu et al., 2024a): A SOTA recommendation loss derived from the listwise MLE, which has been proven as a DCG surrogate loss.
  - **Hyperparameters**: lr $\in \{10^{-1}, 10^{-2}, 10^{-3}\}$, wd $\in \{0, 10^{-4}, 10^{-5}, 10^{-6}\}$, temperature $\tau \in \{0.01, 0.05, 0.1, 0.2, 0.5\}$.
  - **Score function** $s_{ui}$: cosine similarity.

- **AdvInfoNCE** (Zhang et al., 2024): A DRO-based modification of SL. AdvInfoNCE tries to introduce adaptive negative hardness to pairwise score $d_{uij}$ of SL.
  - **Hyperparameters**: lr $\in \{10^{-1}, 10^{-2}, 10^{-3}\}$, wd $\in \{0, 10^{-4}, 10^{-5}, 10^{-6}\}$, temperature $\tau \in \{0.01, 0.05, 0.1, 0.2, 0.5\}$. The other hyperparameters are fixed as the original setting in Zhang et al. (2024). Specifically, the negative weight is set as 64, the adversarial learning will be performed every 5 epochs, with the adversarial learning rate as $5 \times 10^{-5}$.

- **Score function** $s_{ui}$: cosine similarity.
- **BSL** (Wu et al., 2024b): A DRO-based modification of SL. Compared to SL, BSL applies additional DRO on the positive items.
  - **Hyperparameters**: lr $\in \{10^{-1}, 10^{-2}, 10^{-3}\}$, wd $\in \{0, 10^{-4}, 10^{-5}, 10^{-6}\}$, temperatures $\tau_1, \tau_2 \in \{0.01, 0.05, 0.1, 0.2, 0.5\}$.
  - **Score function** $s_{ui}$: cosine similarity.
- **LambdaRank** (Burges et al., 2006): A weighted BPR loss, with weights designed heuristically. LambdaRank aims to optimize DCG, but it is not strictly a DCG surrogate loss.
  - **Hyperparameters**: lr $\in \{10^{-1}, 10^{-2}, 10^{-3}, 10^{-4}\}$, wd $\in \{0, 10^{-4}, 10^{-5}, 10^{-6}\}$.
  - **Score function** $s_{ui}$: dot product.
- **LambdaLoss** (Wang et al., 2018): A DCG@ surrogate loss, which is formally a weighted BPR loss. Wang et al. (2018) finds that LambdaRank does not directly optimize DCG, and proposes LambdaLoss which serves as a DCG surrogate loss.
  - **Hyperparameters**: lr $\in \{10^{-1}, 10^{-2}, 10^{-3}, 10^{-4}\}$, wd $\in \{0, 10^{-4}, 10^{-5}, 10^{-6}\}$.
  - **Score function** $s_{ui}$: dot product.
- **LambdaLoss@$K$** (Jagerman et al., 2022): A DCG@$K$ surrogate loss, which is formally a weighted BPR loss. Based on the LambdaLoss framework, Jagerman et al. (2022) proposes LambdaLoss@$K$ which strictly serves as a DCG@$K$ surrogate loss.
  - **Hyperparameters**: lr $\in \{10^{-1}, 10^{-2}, 10^{-3}, 10^{-4}\}$, wd $\in \{0, 10^{-4}, 10^{-5}, 10^{-6}\}$.
  - **Score function** $s_{ui}$: dot product.
- **SL@$K$ (Ours)**: A DCG@$K$ surrogate loss, which is formally a weighted SL with weight $w_{ui} = \sigma_w(s_{ui} - \hat{\beta}_u^K)$.
  - **Hyperparameters**: lr $\in \{10^{-1}, 10^{-2}, 10^{-3}\}$, wd $\in \{0, 10^{-4}, 10^{-5}, 10^{-6}\}$, SL temperature $\tau_d \in \{0.01, 0.05, 0.1, 0.2, 0.5\}$ (directly using the optimal temperature hyperparameter of SL), weight temperature $\tau_w \in [0.5, 3.0]$ with searching step of 0.25, quantile update interval $T_\beta \in \{5, 20\}$.
  - **Score function** $s_{ui}$: cosine similarity.

### F.5 COMPUTATIONAL RESOURCES

All experiments are conducted on one NVIDIA GeForce RTX 4090 GPU. The code are implemented in PyTorch (Paszke et al., 2019) and will be released upon acceptance.

### F.6 OPTIMAL HYPERPARAMETERS

We report the optimal hyperparameters of each method on each dataset and backbone as the following tables Tables F.4 to F.9, in the order of the hyperparameters listed in Table F.3.

Table F.3: Hyperparameters to be searched for each method.

| Method | Hyperparameters |
|---|---|
| BPR | lr, wd |
| GuidedRec | lr, wd |
| LLPAUC | lr, wd, $\alpha$, $\beta$ |
| SL | lr, wd, $\tau$ |
| AdvInfoNCE | lr, wd, $\tau$ |
| BSL | lr, wd, $\tau_1$, $\tau_2$ |
| LambdaRank | lr, wd |
| LambdaLoss | lr, wd |
| LambdaLoss@$K$ | lr, wd |
| SL@$K$ | lr, wd, $\tau_d$, $\tau_w$, $T_\beta$ |

Table F.4: Optimal hyperparameters of each method on the Health dataset.

| Model | Loss | Hyperparameters | | | | |
|---|---|---|---|---|---|---|
| MF | BPR | 0.001 | 0.0001 | | | |
| | GuidedRec | 0.01 | 0 | | | |
| | LLPAUC | 0.1 | 0 | 0.7 | 0.01 | |
| | SL | 0.1 | 0 | 0.2 | | |
| | AdvInfoNCE | 0.1 | 0 | 0.2 | | |
| | BSL | 0.1 | 0 | 0.2 | 0.2 | |
| | SL@5 | 0.1 | 0 | 0.2 | 2.5 | 20 |
| | SL@20 | 0.1 | 0 | 0.2 | 2.5 | 5 |
| | SL@50 | 0.1 | 0 | 0.2 | 2.5 | 5 |
| LightGCN | BPR | 0.001 | 0.000001 | | | |
| | GuidedRec | 0.01 | 0 | | | |
| | LLPAUC | 0.1 | 0 | 0.7 | 0.1 | |
| | SL | 0.1 | 0 | 0.2 | | |
| | AdvInfoNCE | 0.1 | 0 | 0.2 | | |
| | BSL | 0.1 | 0 | 0.05 | 0.2 | |
| | SL@5 | 0.1 | 0 | 0.2 | 2.5 | 5 |
| | SL@20 | 0.1 | 0 | 0.2 | 2.25 | 20 |
| | SL@50 | 0.1 | 0 | 0.2 | 2.25 | 5 |
| XSimGCL | BPR | 0.1 | 0.000001 | | | |
| | GuidedRec | 0.001 | 0.000001 | | | |
| | LLPAUC | 0.1 | 0 | 0.1 | 0.1 | |
| | SL | 0.1 | 0 | 0.2 | | |
| | AdvInfoNCE | 0.1 | 0 | 0.2 | | |
| | BSL | 0.1 | 0 | 0.05 | 0.2 | |
| | SL@5 | 0.1 | 0 | 0.2 | 1.5 | 5 |
| | SL@20 | 0.1 | 0 | 0.2 | 1.5 | 5 |
| | SL@50 | 0.1 | 0 | 0.2 | 1.5 | 20 |

Table F.5: Optimal hyperparameters of each method on the Electronic dataset.

| Model | Loss | Hyperparameters | | | | |
|---|---|---|---|---|---|---|
| MF | BPR | 0.001 | 0.00001 | | | |
| | GuidedRec | 0.01 | 0 | | | |
| | LLPAUC | 0.1 | 0 | 0.5 | 0.01 | |
| | SL | 0.01 | 0 | 0.2 | | |
| | AdvInfoNCE | 0.1 | 0 | 0.2 | | |
| | BSL | 0.1 | 0 | 0.5 | 0.2 | |
| | SL@5 | 0.1 | 0 | 0.2 | 2.5 | 5 |
| | SL@20 | 0.1 | 0 | 0.2 | 2.25 | 5 |
| | SL@50 | 0.1 | 0 | 0.2 | 2.25 | 20 |
| LightGCN | BPR | 0.01 | 0.000001 | | | |
| | GuidedRec | 0.01 | 0 | | | |
| | LLPAUC | 0.1 | 0 | 0.5 | 0.01 | |
| | SL | 0.01 | 0 | 0.2 | | |
| | AdvInfoNCE | 0.01 | 0 | 0.2 | | |
| | BSL | 0.01 | 0 | 0.2 | 0.2 | |
| | SL@5 | 0.1 | 0 | 0.2 | 2.25 | 5 |
| | SL@20 | 0.1 | 0 | 0.2 | 2.25 | 20 |
| | SL@50 | 0.1 | 0 | 0.2 | 2 | 20 |
| XSimGCL | BPR | 0.01 | 0 | | | |
| | GuidedRec | 0.01 | 0 | | | |
| | LLPAUC | 0.1 | 0 | 0.3 | 0.01 | |
| | SL | 0.01 | 0 | 0.2 | | |
| | AdvInfoNCE | 0.1 | 0 | 0.2 | | |
| | BSL | 0.1 | 0 | 0.1 | 0.2 | |
| | SL@5 | 0.1 | 0 | 0.2 | 1.25 | 20 |
| | SL@20 | 0.1 | 0 | 0.2 | 1.25 | 20 |
| | SL@50 | 0.1 | 0 | 0.2 | 1.25 | 5 |

Table F.6: Optimal hyperparameters of each method on the Gowalla dataset.

| Model | Loss | Hyperparameters | | | | |
|---|---|---|---|---|---|---|
| MF | BPR | 0.001 | 0.000001 | | | |
| | GuidedRec | 0.001 | 0 | | | |
| | LLPAUC | 0.1 | 0 | 0.7 | 0.01 | |
| | SL | 0.1 | 0 | 0.1 | | |
| | AdvInfoNCE | 0.1 | 0 | 0.1 | | |
| | BSL | 0.1 | 0 | 0.2 | 0.1 | |
| | SL@5 | 0.1 | 0 | 0.1 | 1 | 20 |
| | SL@20 | 0.1 | 0 | 0.1 | 1 | 20 |
| | SL@50 | 0.1 | 0 | 0.1 | 1 | 20 |
| LightGCN | BPR | 0.001 | 0 | | | |
| | GuidedRec | 0.001 | 0 | | | |
| | LLPAUC | 0.1 | 0 | 0.7 | 0.01 | |
| | SL | 0.1 | 0 | 0.1 | | |
| | AdvInfoNCE | 0.1 | 0 | 0.1 | | |
| | BSL | 0.1 | 0 | 0.05 | 0.1 | |
| | SL@5 | 0.1 | 0 | 0.1 | 0.75 | 5 |
| | SL@20 | 0.1 | 0 | 0.1 | 0.75 | 5 |
| | SL@50 | 0.1 | 0 | 0.1 | 0.75 | 5 |
| XSimGCL | BPR | 0.0001 | 0 | | | |
| | GuidedRec | 0.001 | 0 | | | |
| | LLPAUC | 0.1 | 0 | 0.7 | 0.01 | |
| | SL | 0.01 | 0 | 0.1 | | |
| | AdvInfoNCE | 0.1 | 0 | 0.1 | | |
| | BSL | 0.1 | 0 | 0.05 | 0.1 | |
| | SL@5 | 0.1 | 0 | 0.1 | 0.75 | 20 |
| | SL@20 | 0.1 | 0 | 0.1 | 0.75 | 5 |
| | SL@50 | 0.1 | 0 | 0.1 | 0.75 | 5 |

Table F.7: Optimal hyperparameters of each method on the Book dataset.

| Model | Loss | Hyperparameters | | | | |
|---|---|---|---|---|---|---|
| MF | BPR | 0.0001 | 0 | | | |
| | GuidedRec | 0.001 | 0 | | | |
| | LLPAUC | 0.1 | 0 | 0.7 | 0.01 | |
| | SL | 0.1 | 0 | 0.05 | | |
| | AdvInfoNCE | 0.01 | 0 | 0.1 | | |
| | BSL | 0.1 | 0 | 0.5 | 0.05 | |
| | SL@5 | 0.1 | 0 | 0.05 | 0.5 | 5 |
| | SL@20 | 0.1 | 0 | 0.05 | 0.5 | 20 |
| | SL@50 | 0.1 | 0 | 0.05 | 0.5 | 5 |
| LightGCN | BPR | 0.001 | 0 | | | |
| | GuidedRec | 0.001 | 0 | | | |
| | LLPAUC | 0.1 | 0 | 0.7 | 0.01 | |
| | SL | 0.1 | 0 | 0.05 | | |
| | AdvInfoNCE | 0.1 | 0 | 0.1 | | |
| | BSL | 0.1 | 0 | 0.5 | 0.05 | |
| | SL@5 | 0.1 | 0 | 0.05 | 0.5 | 20 |
| | SL@20 | 0.1 | 0 | 0.05 | 0.5 | 20 |
| | SL@50 | 0.1 | 0 | 0.05 | 0.5 | 20 |
| XSimGCL | BPR | 0.0001 | 0.00001 | | | |
| | GuidedRec | 0.1 | 0 | | | |
| | LLPAUC | 0.1 | 0 | 0.7 | 0.01 | |
| | SL | 0.1 | 0 | 0.05 | | |
| | AdvInfoNCE | 0.1 | 0 | 0.1 | | |
| | BSL | 0.1 | 0 | 0.05 | 0.05 | |
| | SL@5 | 0.1 | 0 | 0.05 | 0.5 | 20 |
| | SL@20 | 0.1 | 0 | 0.05 | 0.5 | 20 |
| | SL@50 | 0.1 | 0 | 0.05 | 0.5 | 20 |

Table F.8: Optimal hyperparameters of each method on the Movielens dataset.

| Model | Loss | Hyperparameters | | | | |
|---|---|---|---|---|---|---|
| MF | LambdaRank | 0.01 | 0.000001 | | | |
| | LambdaLoss | 0.001 | 0.00001 | | | |
| | LambdaLoss (Sample) | 0.01 | 0.0001 | | | |
| | LambdaLoss@20 | 0.001 | 0.00001 | | | |
| | LambdaLoss@20 (Sample) | 0.01 | 0.00001 | | | |
| | SL@20 | 0.1 | 0 | 0.2 | 3 | 5 |

Table F.9: Optimal hyperparameters of each method on the Food dataset.

| Model | Loss | Hyperparameters | | | | |
|---|---|---|---|---|---|---|
| MF | LambdaRank | 0.001 | 0.00001 | | | |
| | LambdaLoss | 0.01 | 0.00001 | | | |
| | LambdaLoss (Sample) | 0.001 | 0.0001 | | | |
| | LambdaLoss@20 | 0.001 | 0.00001 | | | |
| | LambdaLoss@20 (Sample) | 0.01 | 0.000001 | | | |
| | SL@20 | 0.1 | 0 | 0.2 | 2.25 | 5 |

## G   SUPPLEMENTARY EXPERIMENTAL RESULTS

### G.1   SUPPLEMENTARY RESULTS: SL@$K$ VS. LAMBDALOSS@$K$

Supplementary results of Table 3 are reported in Table G.10. We compare the performance of SL@$K$ with three Lambda losses, including LambdaRank (Burges et al., 2006), LambdaLoss (Wang et al., 2018), and LambdaLoss@$K$ (Jagerman et al., 2022).

Table G.10: Supplementary results of Table 3. Performance comparison of SL@$K$ with Lambda losses on MF backbone, including LambdaRank, LambdaLoss, and LambdaLoss@$K$. The best results are highlighted in bold, and the best baselines are underlined. "Imp." denotes the improvement of SL@$K$ over the best Lambda loss, while "Degr." denotes the degradation of Lambda losses caused by the sample ranking estimation (cf. Appendix D.4).

| Loss | Movielens | | Food | |
|---|---|---|---|---|
| | **Recall@20** | **NDCG@20** | **Recall@20** | **NDCG@20** |
| LambdaRank | 0.3077 | 0.3043 | 0.0520 | 0.0377 |
| LambdaLoss | 0.3425 | 0.3460 | 0.0515 | 0.0374 |
| LambdaLoss (Sample) | 0.1497 | 0.1523 | 0.0333 | 0.0243 |
| Degr. (Sample) % | -56.29% | -55.98% | -35.34% | -35.03% |
| LambdaLoss@20 | 0.3418 | 0.3466 | 0.0530 | 0.0382 |
| LambdaLoss@20 (Sample) | 0.1580 | 0.1603 | 0.0335 | 0.0238 |
| Degr. (Sample) % | -53.77% | -53.75% | -36.79% | -37.70% |
| **SL@20** | **0.3580** | **0.3677** | **0.0635** | **0.0465** |
| **Imp. %** | **+4.53%** | **+6.09%** | **+19.81%** | **+21.73%** |

### G.2   SUPPLEMENTARY RESULTS: NOISE ROBUSTNESS STUDY

Supplementary results of Figure 3 are reported in Figure G.4. We compare the performance of SL@$K$ with SL and DRO-based BSL under False Positive Noise scenario with varying ratios of imposed false positive instances.

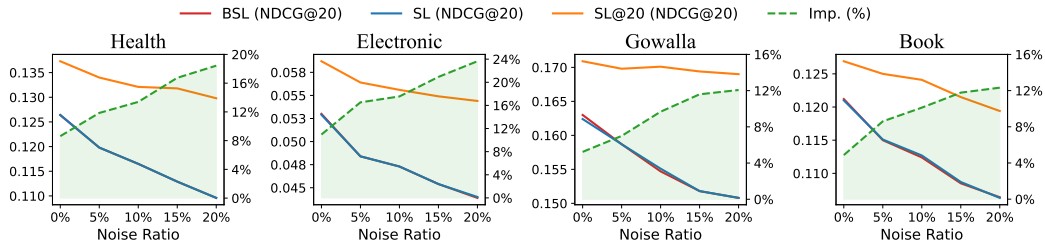

Figure G.4: Supplementary results of Figure 3. NDCG@20 Performance of SL@$K$ compared with SL and BSL under varying ratios of imposed false positive instances. "Imp." indicates the improvement of SL@$K$ over SL.

