# OpenReview forum: "Towards Optimizing Top-$K$ Ranking Metrics in Recommender Systems"
_ICLR.cc/2025/Conference — ICLR 2025 Conference Withdrawn Submission_

### Official Review · Reviewer_t9b3 · 2024-10-31

**Soundness:** 3
**Presentation:** 3
**Contribution:** 2
**Rating:** 5
**Confidence:** 5

**Summary:**

Recommender systems are often referred to as top-K item recommendation tasks, and as such, much of the literature over the past decade has sought solutions to optimize these top-K ranking metrics. This paper makes a similar argument and proposes a new ranking loss called SoftmaxLoss. The authors provide a reasonable analysis of the difficulty of optimizing NDCG and derive a mathematical derivation of how NDCG might be optimized. In particular, the authors make multiple approximations in Equations 3.3b, 3.3d, 3.3e. Finally, in Equation 3.5, they derive the so-called loss. They perform extensive experiments to show that SoftmaxLoss is a good optimizer for top-N ranking.

**Strengths:**

(1) The paper is well written and easy to understand, with clear motivation and ideas

(2) The authors provide a rationale for the NDCG optimization problem

(3) A large number of experiments are performed to support the claim

**Weaknesses:**

(1) While I agree that optimizing top-K metrics such as NDCG or MRR is important for recommendation tasks, the existing literature over the past decade is extensive. Although numerous new solutions have been proposed, the commonly used loss functions still primarily include cross-entropy loss, batch softmax loss [1], and negative sampling-based pairwise ranking [2,3,5]. In my view, this particular solution does not significantly advance the field, as the ranking problem has been well-established in recent years; it is a relatively old topic.

(2) The proposed solution primarily addresses the recall stage, while CTR prediction tasks continue to focus on AUC during the offline phase. The relative improvements in the recall stage may not have a substantial impact on the final ranking stage.

(3) As I mentioned, there is an abundance of related literature, and the authors may have overlooked some key works. For example, the authors compare their approach to BPR loss, which is a well-known baseline function. Many studies have indicated that incorporating negative sampling can significantly enhance BPR's performance. The authors should consider comparing their results with other relevant literature, such as WARP loss[2], LambdaFM loss[3], and batch softmax loss[1].

(4) Additionally, while the topic falls under the realm of learning to rank, it is noteworthy that learning-to-rank has not been prominently featured in recent literature. The 2011 Yahoo Learning to Rank Challenge highlighted that optimization of top-K ranking metrics is not particularly impressive compared to classical regression and classification losses.

All in all, the topic discussed in this article is not really interesting to me personally and it does not address a very important problem in the field of current recommender systems. It feels like just another paper on the subject.


[1] Sampling-bias-corrected neural modeling for large corpus item recommendations. Recsys2019

[2] WSABIE: Scaling Up To Large Vocabulary Image Annotation. IJCAI 2011

[3] LambdaFM: Learning Optimal Ranking with Factorization Machines Using Lambda Surrogates. CIKM2016

[4] Yahoo! learning to rank challenge overview. Proceedings of the learning to rank challenge 2011

[5] Co-Factorization Machines: Modeling User Interests and Predicting Individual Decisions in Twitter.WSDM 2013

**Questions:**

No

---

> ### Author Response · Authors · 2024-11-15
> **Response to Reviewer t9b3**
>
> > **W1.** Concerns on the importance of the study on the recommendation loss functions.
>
> **R1.** While recommendation loss has been studied for over 10 years, there are still open problems, and it remains a hot research topic. We have observed a surge of work in the last year, e.g., Softmax Loss (SL) (TOIS '24) [1], BSL (ICDE '24) [2], AdvInfoNCE (NIPS '24) [3], LLPAUC (WWW '24) [4], to name a few. To the best of our knowledge, some of these have been applied in industry and have indeed shown improvements.
>
> This work addresses a less explored problem: how to optimize the Top-$K$ metric in recommendation. The proposed loss indeed brings significant improvements over multiple real-world datasets and recommendation backbones.
>
> > **W2.** The proposed solution primarily addresses the recall stage. The relative improvements in the recall stage may not have a substantial impact on the final ranking stage.
>
> **R2.** To the best of our knowledge, and discussed with the industry engineers, it seems that the recall stage is also highly important. Its quality determines the upper bound of the recommendation performance.  Moreover, the proposed loss can also be utilized in the fine-ranking stage, serving as a part of multi-objective optimization, given that only top-$K$ items are shown to users.
>
> > **W3.** Concerns on missing compared methods and references.
>
> **R3.** Thanks for your reference. We will include these citations in the next version. However, these losses are relatively outdated and perform worse than recent advances, e.g., Softmax Loss (SL) (TOIS '24) [1], BSL (ICDE '24) [2], AdvInfoNCE (NIPS '24) [3], LLPAUC (WWW '24) [4], which have been compared in our experiments.
>
> > **W4.** It is noteworthy that learning-to-rank has not been prominently featured in recent literature. The 2011 Yahoo Learning to Rank Challenge highlighted that optimization of top-K ranking metrics is not particularly impressive compared to classical regression and classification losses.
>
> **R4.** Thanks for the insightful comments. Yes, optimization of Top-$K$ ranking metrics is not particularly impressive at the 2011 Yahoo Learning to Rank Challenge. However, it should be noted that over one decade has passed, the architecture of recommendation models and the scale of datasets are changing. With the development of deep learning, embedding-based recommendation methods have become mainstream, replacing earlier linear or tree-based methods. Additionally, the scale of users and items is increasing rapidly, making losses like LambdaRank [5] and LambdaLoss [6] impractical. These changes drive new studies of loss functions.
>
> Moreover, the poor performance of existing Top-$K$ losses does not suggest that optimizing Top-$K$ metrics is unimportant or ineffective. In fact, we have also tested recent work on top-$K$ metrics optimization, like LambdaLoss@$K$ [7], which sometimes performs worse than basic SL or BPR. These losses have certain limitations, and we need to develop better Top-$K$ metrics surrogate loss. With careful design, our SL@$K$ achieves much better performance than existing losses.
>
> **Reference:**
>
> - [1] On the effectiveness of sampled softmax loss for item recommendation. TOIS 2024.
> - [2] BSL: Understanding and improving softmax loss for recommendation. ICDE 2024.
> - [3] Empowering Collaborative Filtering with Principled Adversarial Contrastive Loss. NIPS 2024.
> - [4] Lower-Left Partial AUC: An Effective and Efficient Optimization Metric for Recommendation. WWW 2024.
> - [5] Learning to rank with nonsmooth cost functions. NIPS 2006.
> - [6] The lambdaloss framework for ranking metric optimization. CIKM 2018.
> - [7] On optimizing top-k metrics for neural ranking models. SIGIR 2022.

---

### Official Review · Reviewer_4CMb · 2024-11-02

**Soundness:** 3
**Presentation:** 3
**Contribution:** 2
**Rating:** 5
**Confidence:** 4

**Summary:**

In this paper, the authors propose a new softmax loss as a surrogate for optimizing NDCG@K in ranking tasks. My primary concerns revolve around the paper's motivation and experimental validation.

**Strengths:**

1. The topic of this paper is highly interesting.
2. The paper provides a theoretical guarantee for optimizing the proposed method.
3. The proposed methods demonstrate significant improvements in experiments.

**Weaknesses:**

1. The motivation of this paper is problematic.
2. The softmax loss is not novel; therefore, the paper should clarify its novelty and distinguish the proposed methods from existing solutions.
3. The experiments only include small datasets and traditional ranking methods. I highly recommend that the authors use large-scale recommendation datasets and incorporate modern recommendation and ranking models.

**Questions:**

The authors’ exploration of ranking loss is compelling and relevant to real-world applications. My concerns center on the following aspects. Firstly, the inconsistency between NDCG and NDCG@K appears similar to the difference between NDCG@K1 and NDCG@K2 for different values of K. However, this observation alone does not provide a basis for questioning the NDCG metric itself, as NDCG@K is designed to evaluate ranking performance for the top-K items, which naturally varies with different values of K. I highly encourage the authors to clarify their motivation further. Additionally, if they wish to highlight this observation, it would be useful to evaluate the inconsistency between SL@K1 and SL@K2 as well.

I appreciate the authors' effort in introducing a new loss function, and I suggest they summarize the distinctions between their proposed loss function and existing softmax loss functions [1]. In the experiments, I recommend including models such as DeepFM [2], DIN [3], and SIM [4], as MF and LightGCN are less common in real-world recommendation settings.

[1] On the Effectiveness of Sampled Softmax Loss for Item Recommendation
[2] DeepFM: A Factorization-Machine based Neural Network for CTR Prediction
[3] Deep Interest Network for Click-Through Rate Prediction
[4] Search-based User Interest Modeling with Lifelong Sequential Behavior Data for Click-Through Rate Prediction

---

> ### Author Response · Authors · 2024-11-15
> **Response to Reviewer 4CMb**
>
> > **Q1.** Concerns on the motivaition: the inconsistency between NDCG and NDCG@K appears similar to the difference between NDCG@K1 and NDCG@K2 for different values of K.
>
> **R1.** Thank you for the insightful comments. Yes, different values of $K$ in NDCG@$K$ can lead to the metrics inconsistencies. In other words, NDCG@1 differs from NDCG@2. In fact, we have conducted experiments to validate this phenomenon, as shown in Table 4. We observe the different recommendation performance in terms of different NDCG@$K$ when optimizing the model with different SL@$K$. It motivates us to leverage concrete Top-$K$ truncation in model optimization.
>
> > **Q2.** Suggestion: summarize the distinctions between their proposed loss function and existing softmax loss functions.
>
> **R2.** Thanks for raising this concern. We have discussed the connections and differences between the proposed SL@$K$ with SL in the page 6 lines 304-309: "Compared to conventional SL (Softmax Loss), SL@$K$ only introduces an additional quantile-based weight $w_{ui}$ for each instance, which just involves a simple difference between the scores $s_{ui}$ and the quantile $\beta^K_u$." This simple difference makes SL@$K$ serves as a tight surrogate for NDCG@$K$, while SL does not.

---

### Official Review · Reviewer_ESxR · 2024-11-02

**Soundness:** 3
**Presentation:** 3
**Contribution:** 2
**Rating:** 3
**Confidence:** 4

**Summary:**

The paper introduces a novel ranking loss function that optimises ndcg@k a popular evaluation metric utilised in recommender systems. The authors base the loss function on a sampled version of the softmax loss version where a quantile technique is used to separate the items in the top K from the rest. The authors provide a comparison to other similar loss functions and a fairly extensive experimental section which demonstrates significant gains compared to other loss functions. '
The related work, analysis of the lambda loss function and proofs are included in the appendix.

**Strengths:**

The paper is well written and easy to follow for the most part.
The loss function can be of potential practical use in some recommender systems applications.
The experimental section is fairly extensive.

**Weaknesses:**

The contribution of the paper is rather limited, over the last 10 years a large number of ranking loss functions have been proposed.
The core topic of this paper falls somewhat outside of the core interests of this conference, a IR or recommender systems conference might be more appropriate.
Code is not included in the submission
It is unclear how this loss function would perform compared to the large number of already proposed loss function for recommendation.
While optimising for IR evaluation metrics is a good way of showing increases in offline experimental results in recommender systems papers it is unclear how relevant these gains are in real online recommendation systems as techniques as off-policy correction and IPS seem to produce bigger gains in real world recommendation engines rather than then latest ranking loss function.

**Questions:**

My main suggestion is to find a more appropriate venue for this paper that is closer to the core audience that could be interested in the topic.

---

> ### Author Response · Authors · 2024-11-15
> **Response to Reviewer ESxR**
>
> > **W1.** The contribution of the paper is rather limited, over the last 10 years a large number of ranking loss functions have been proposed. While optimising for IR evaluation metrics is a good way of showing increases in offline experimental results in recommender systems papers it is unclear how relevant these gains are in real online recommendation systems as techniques as off-policy correction and IPS seem to produce bigger gains in real world recommendation engines rather than then latest ranking loss function.
>
> **R1.** While recommendation loss has been studied for over 10 years, there are still open problems, and it remains a hot research topic. We have observed a surge of work in the last year, e.g., Softmax Loss (SL) (TOIS '24) [1], BSL (ICDE '24) [2], AdvInfoNCE (NIPS '24) [3], LLPAUC (WWW '24) [4], to name a few. To the best of our knowledge, some of these have been applied in industry and have indeed shown improvements. Additionally, we totally agree that addressing bias is important in the recommendation area, but this does not suggest that the loss function is unimportant.
>
> > **W2.** Code is not included in the submission.
>
> **R2.** We had provided the code in the supplementary material in the initial submission.
>
> **Reference:**
>
> - [1] On the effectiveness of sampled softmax loss for item recommendation. TOIS 2024.
> - [2] BSL: Understanding and improving softmax loss for recommendation. ICDE 2024.
> - [3] Empowering Collaborative Filtering with Principled Adversarial Contrastive Loss. NIPS 2024.
> - [4] Lower-Left Partial AUC: An Effective and Efficient Optimization Metric for Recommendation. WWW 2024.

---

### Official Review · Reviewer_MQaV · 2024-11-05

**Soundness:** 2
**Presentation:** 3
**Contribution:** 2
**Rating:** 3
**Confidence:** 4

**Summary:**

Summary:

This paper proposes a new loss function named SoftmaxLoss@K (SL@K), which could be an alternative choice when it comes to optimizing NDCG@K for recommender system.

**Strengths:**

Strengths:

-	The idea is simple and intuitive, easy to understand.
-	The experiments are conducted based on well-known datasets and baselines (and algorithms)
-	The authors also provided ablation studies
-	The authors provided source code for reproducibility and details in the Appendix are also given.

**Weaknesses:**

Weaknesses:

-	In my opinion, the contribution is marginal, as compared with previous works in [1, 2, 3].
-	It’s still not very clear about the benefits of using SL@K. Simply looking at Table 1, 2, 3, it appears that the best baseline performance is not stable across scenarios, is it an advantage of SL@K for stable / trustable performance?
-	Since the work focuses on SL@K; it’s worth to explore more numbers of K (e.g., 5,10,15,20,25,…100) instead of just a few Ks.
-	As a follow-up, it’s better to compare SL@K, NDCG@K, and LambdaLoss@K across experiments (e.g., Figure 3 and Table 4)
-	If the authors would like to emphasize on the practical applications (as compared to LambdaLoss@K for example), more examples should be given with large dataset such as Netflix or MovieLens20M. I understand Table 3 is given, but in my opinion, small datasets with MF backbone cannot guaranteed its practical application for large-scale RS (as mentioned in L183-200

**Questions:**

Please refer to the weaknesses section; I may have more questions later.

---

> ### Author Response · Authors · 2024-11-15
> **Response to Reviewer MQaV**
>
> > **W1.** The contribution is marginal, as compared with previous works in [1, 2, 3].
>
> **R1.** It seems that the mentioned references "[1, 2, 3]" have been omitted in the comments. We cannot provide detailed responses for clarification.
>
> > **W2.** It appears that the best baseline performance is not stable across scenarios. Is it an advantage of SL@K for stable / trustable performance?
>
> **R2.**  There seems to be a misunderstanding. From Tables 1, 2, and 3, we can observe that the improvements of SL@$K$ over the best baselines are quite significant, with an average improvement of 6.19%.
>
> > **W3.** It’s worth to explore more numbers of K (e.g., 5, 10, 15, 20, 25, ..., 100) instead of just a few Ks.
>
> **R3.** Thanks for your suggestion. We have tested $K \in [5, 20, 50]$, and we will examine more value of $K$ in future versions.
>
> > **W4.** As a follow-up, it’s better to compare SL@K, NDCG@K, and LambdaLoss@K across experiments (e.g., Figure 3 and Table 4)
>
> **R4.** We tested LambdaLoss@$K$ on two relatively small datasets due to its high time complexity (cf. Section 2.3), which makes it unsuitable for larger datasets, such as Gowalla or Book datasets. For NDCG@$K$, it seems it is an evaluation metric rather than a loss function.
>
> > **W5.** If the authors would like to emphasize on the practical applications (as compared to LambdaLoss@K for example), more examples should be given with large dataset such as Netflix or MovieLens20M.
>
> **R5.** Thanks for your suggested datasets. We understand your concerns. In fact, the number of users in the Book dataset we used is nearly the same as the Movielens-20M dataset, and the number of items is significantly larger than Movielens-20M (10 times). Additionally, we chose to use these Amazon and Gowalla datasets as they are more commonly used in recent studies on loss functions than Movielens-20M and Netflix.

---

### Author Response · Authors · 2024-11-14
**Overall Clarifications (Part 1/2)**

We thank the reviewers for their efforts and detailed comments. We are disappointed that the contributions of this paper, the problem it addresses, and even the entire field of recommendation loss, have been misunderstood or neglected. It appears that this manuscript may not be suitable for this venue, and we have decided to withdraw our submission.

Below are some overall clarifications of our work:

**C1. Importance of Recommendation Loss.**

While recommendation loss has been studied for over 10 years, there are still open problems, and it remains a hot research topic. We have observed a surge of work in the last year, e.g., Softmax Loss (SL) (TOIS '24) [1], BSL (ICDE '24) [2], AdvInfoNCE (NIPS '24) [3], LLPAUC (WWW '24) [4], to name a few. To the best of our knowledge, some of these have been applied in industry and have indeed shown improvements.

**C2. Importance of the problem we address.**

Our goal is to design an new loss that targets at optimizing the NDCG@$K$ metric, which is commonly used in recommender systems but is less explored in this area. NDCG@$K$ differs from these global metrics (e.g., NDCG, Recall, or AUC) as it involves a Top-$K$ truncation. The NDCG@$K$ performance just involves the items within the Top-$K$ ranking positions. Optimizing global metrics may not yield consistent improvements on NDCG@$K$ (cf. Figure 1a). Such inconsistence motivates us to explore a new surrogate loss of NDCG@$K$.

**C3. Our contributions.**

Through rigorous theoretical derivation, we have developed a new NDCG@$K$ surrogate loss --- **SoftmaxLoss@$K$ (SL@$K$)**. Optimizing NDCG@$K$ is challenging, and so far, only LambdaLoss@$K$ [5] provides a strict NDCG@$K$ surrogate loss, while other methods overlook the truncation and just optimizes the global metric NDCG.

In contrast, our SL@$K$ loss has the following advantages:

- **Effective**: Directly optimizes for NDCG@$K$ with theoretical accuracy guarantees (cf. Section 3.1), obtaining impressive performance improvements (avg. 6.19%) over SOTA methods (cf. Table 1).
- **Efficient**: SL@$K$ has similar complexity to SL, offering higher efficiency, especially compared to LambdaLoss@K with quadratic complexity (cf. Section 3.2 and Table 3).
- **Easy and Flexible**: Formulated as a simple weighted SL, applicable in any scenario where SL can be used, requiring minimal code changes.

**Reference:**

- [1] On the effectiveness of sampled softmax loss for item recommendation. TOIS 2024.
- [2] BSL: Understanding and improving softmax loss for recommendation. ICDE 2024.
- [3] Empowering Collaborative Filtering with Principled Adversarial Contrastive Loss. NIPS 2024.
- [4] Lower-Left Partial AUC: An Effective and Efficient Optimization Metric for Recommendation. WWW 2024.
- [5] On optimizing top-k metrics for neural ranking models. SIGIR 2022.

---

> ### Author Response · Authors · 2024-11-14
> **Overall Clarifications (Part 2/2)**
>
> Thanks to all the reviewers for their detailed comments. Some reviewers have raised concerns about this work. While we have decided to withdraw the paper, we would still like to provide some necessary clarifications. We hope these will help the reviewers better understand this paper and address their concerns.
>
> Lastly, thank you again for your time and effort in reviewing our paper.

---

### Note · Authors · 2024-11-21

I have read and agree with the venue's withdrawal policy on behalf of myself and my co-authors.